# Development of a Cyberinfrastructure for Assessment of the Lower Rio Grande Valley North and Central Watersheds Characteristics

**Linda Navarro [1], Ahmed Mahmoud [2,\*], Andrew Ernest [1], Abdoul Oubeidillah [1], Jessica Johnstone [3], Ivan Rene Santos Chavez [1] and Christopher Fuller [4]**

1  Department of Civil Engineering, University of Texas Rio Grande Valley, Edinburg, TX 78539, USA; lnavarro@office.ratesresearch.org (L.N.); andrew.ernest@utrgv.edu (A.E.); abdoul.oubeidillah@utrgv.edu (A.O.); ivan.santoschavez01@utrgv.edu (I.R.S.C.)
2  Department of Biological and Agricultural Engineering, University of Arkansas, Fayetteville, AR 72704, USA
3  Nonpoint Source Program, Texas Commission on Environmental Quality, Austin, TX 78753, USA; jessica.johnstone@tceq.texas.gov
4  Research, Applied, Technology, Education and Service, Inc., Rio Grande Valley, Edinburg, TX 78540, USA; cfuller@office.ratesresearch.org
\*  Correspondence: ahmedm@uark.edu

**Abstract:** Lower Laguna Madre (LLM) is designated as an impaired waterway for high concentrations of bacteria and low dissolved oxygen. The main freshwater sources to the LLM flow from the North and Central waterways which are composed of three main waterways: Hidalgo/Willacy Main Drain (HWMD), Raymondville Drain (RVD), and International Boundary & Water Commission North Floodway (IBWCNF) that are not fully characterized. The objective of this study is to perform a watershed characterization to determine the potential pollution sources of each watershed. The watershed characterization was achieved by developing a cyberinfrastructure, and it collects a wide inventory of data to identify which one of the three waterways has a major contribution to the LLM. Cyberinfrastructure development using the Geographic Information System (GIS) database helped to comprehend the major characteristics of each area contributing to the watershed supported by the analysis of the data collected. The watershed characterization process started with delineating the boundaries of each watershed. Then, geospatial and non-geospatial data were added to the cyberinfrastructure from numerous sources including point and nonpoint sources of pollution. Results showed that HWMD and IBWCNF watersheds were found to have a higher contribution to the water impairments to the LLM. HWMD and IBWCNF comprise the potential major sources of water quality impairments such as cultivated crops, urbanized areas, on-site sewage facilities, colonias, and wastewater effluents.

**Keywords:** watershed management; nonpoint source pollution; point source pollution; water quality; pollutant loadings; South Texas

## 1. Introduction

The Lower Rio Grande Valley (LRGV) region has undergone sudden hydrologic change due to urbanization. This abrupt change has produced a decline in water quality in the primary waterways of the region. The Laguna Madre is an estuarine wetland system along the Gulf of Mexico that receives freshwater from the LRGV [1]. This watershed is known for its recreational activities and is currently threatened by the inflows of main drainage pathways that carry significant levels of contaminants. According to the Texas Commission on Environmental Quality (TCEQ) 2020 Integrated Report [2], two water segments from the Lower Laguna Madre are considered impaired due to high levels of bacteria and low dissolved oxygen. The watershed is comprised of three waterways, Hidalgo/Willacy Main Drain (HWMD), Raymondville Drain (RVD), and International

Boundary & Water Commission North Floodway (IBWCNF), that provide freshwater inflows to the Lower Laguna Madre. Prior to this study, these waterways had not been characterized. Watershed characterization can enable proper identification of potential sources of pollution to help reduce water impairments to the Laguna Madre and preserve the ecosystem.

One of the emerging tools for watershed characterization is cyberinfrastructure that can assist in both data collection and decision-making processes within the watershed. Cyberinfrastructure supports the process of accessing data via an extensive network and provides updated water quality data for further research. The introduction of a cyberinfrastructure can provide an efficient data collection to well demonstrate the watershed characteristics. In one study, cyberinfrastructure not only utilized widespread data but also allowed researchers to analyze large amounts of data over time at different locations [3–5]. This platform offers a rapid generation of new relationships between wide inventories of data. Cyberinfrastructure secures data and delivers interpreted information via a sequence of web services and portals in forms that are universally coherent by distinct stakeholders [6]. Further, it serves as the center for a variety of data from distinct sources, such as non-point and point source and watershed delineation characteristics. Cyberinfrastructure and the watershed delineation are crucial for the watershed characterization since together they will help identify sources of pollution data within the drainage area.

An ample watershed delineation is key for a successful watershed characterization. A watershed delineation is developed by using elevation data and computing several elevation-based files that represent the overall drainage area as well as the hydrological characteristics of a watershed [7]. Each watershed can be divided into sub-watersheds to produce a more detailed drainage structure. The Geographical Information systems (GIS) platform has facilitated the development of hydrological analysis, such as drainage areas based on elevation data. In 2010, a watershed GIS-based applications study performed a hydrological analysis which showed positive outcomes regarding GIS-applications for watershed management and water quality by providing a full overview of watershed characteristics, such as land cover [8]. Hydraulic and hydrological modelling as well as water resource management commonly require investigation of landscape and hydrological features, such as terrain slope, drainage networks, drainage divides, and catchment boundaries [9]. Additionally, high resolution in data resources is important to obtain accurate results in watershed drainage areas [10]. When the land slope is very flat and has few contours, it is challenging for the acquisition of topographic maps. Light Detection and Ranging (LIDAR) is a high-resolution digital elevation model (DEM) that is an ideal source for the type of topography characterized in low elevation areas [11]. Although the terrain in the LRGV is flat, the complex hydrologic features make the process difficult and challenging with even high-resolution DEM. Hence, a previous study focused on enhancing streamlines and watershed boundaries derived from a high-resolution DEM for future hydrologic modeling and flood forecasting [12]. To determine accurate stream networks, an effective method of eliminating pits or depressions is the stream burning algorithm. This algorithm often identifies river channels or lakes that are not recorded in the DEM, avoiding serious errors in the streaming [3,4]. A stream-burning algorithm can enhance the replication of streams' positions by using raster representation of a vector stream network to trench known hydrological features into a DEM, resulting in a comprehensive watershed delineation [4,13,14]. In addition, delineation of watersheds will not only serve to determine drainage boundaries but to distinguish existing sources of nonpoint sources (NPS) and point sources (PS) pollution.

Part of watershed characterization is to identify potential sources of pollution within the watershed. Pollutant sources have been divided into two different classifications: NPS and PS; with this distinction, it becomes easier to study, analyze, understand, and propose actions to mitigate the pollutant load. NPS pollutants are difficult to identify because they cannot be tracked and usually come from several land uses. The major contributor of NPS pollution is stormwater runoff originated by rainfall [15] and other forms of water

flow through several different land uses. They ultimately discharge to lakes, canals, and coastal waters. This runoff carries significant levels of pollution caused by fertilizers, oil, grease, sediments, bacteria, and nutrients [16]. NPS pollutants contained a significant amount of nutrients, such as nitrogen and phosphorus [17]. There has been increasing emphasis on tackling NPS pollution from agricultural land for the presence of high nutrient contamination [18]. Currently, urbanization has led to increased water transfers from agriculture to urban uses [1,19]. These changes are altering the nature, location, and scope of wastewater loadings into the river. Urban runoff has caused negative results on water quality due to high bacteria and low dissolved oxygen (DO) levels [15]. Recent reports indicated that more than 40% of all impaired waters were affected solely by NPS pollutants, while only 10% of impairments were caused by PS pollutant discharges alone [20].

Unlike NPS pollutants, PS pollutants can be identified because they come from only one source. However, they still present a problem when addressing the pollution issues in primary waterways. To establish the proper actions to reduce or stop the pollutant load into waterbodies, it is necessary to identify the source of the pollutant. PS pollution identification is a challenging task because of the uncertainties and nonlinearity in the transport process of pollutants [21]. The typical way to identify PS pollution requires obtaining prior information of the pollution source, gaining complex information about pollution such as incidents regarding flow simulation dimensions, tabulating the number of PS pollutants involved, and evaluating the pollutant release process [22]. Determining potential sources is the first step in acting toward reducing the effects of water quality problems. Unlike NPS, PSs can be identified because they come from only one source. However, they still present a problem when addressing the pollution issues in primary waterways. To establish the proper actions to reduce or stop the pollutant load into waterbodies, it is necessary to identify the source of the pollutant. PS identification is a challenging task because of the uncertainties and nonlinearity in the transport process of pollutants [21]. The typical way to identify a PS requires obtaining prior information of the pollution source, gaining complex information about pollution such as incidents regarding flow simulation dimensions, tabulating the number of PS involved, and evaluating the pollutant release process [22]. Determining potential sources is the first step in acting toward reducing the effects of water quality problems.

Almost 70% of all rivers and streams in the United States are unassessed. In the State of Texas, 88% of all rivers and streams are unassessed. In the United States, 53% of the assessed water bodies were considered impaired due to high levels of *E. coli* and fecal coliform [23]. In addition, fecal coliform bacteria and other pathogens present in stormwater discharges threaten public health and have been responsible for numerous beach closings in the region [24]. Some studies have found that both livestock and manure management can potentially be agricultural sources of fecal indicator bacteria in watersheds [25]. Moreover, estuaries have faced eutrophication because of increased inputs of nutrients, such as nitrogen and phosphorus. This phenomenon is now considered to be a worldwide issue [26–28]. Ammonia can enter the aquatic environment via direct means of municipal effluent discharge and excretion of nitrogenous wastes from animals. It may also contaminate certain areas through indirect means such as nitrogen fixation, air deposition, and runoff from agricultural lands [29]. Improper wastewater management practices in this under-served region have caused severe water quality problems, and sections of the river have experienced poor water quality with regard to dissolved oxygen, bacteria, and algae [30].

The Laguna Madre is identified as an impaired waterbody due to the presence of high concentrations of bacteria and low dissolved oxygen [2]. The Lower Laguna Madre receives freshwater inflows from three waterways located in the north and central part of the LRGV. The three waterways are HWMD, RVD and IBWCNF, which are not fully characterized due to insufficient data. The aim of this paper is to provide a comprehensive characterization of the north and central watersheds to analyze pollution sources. A cyber-infrastructure database was developed to facilitate navigating through distinct information

to obtain potential sources of pollution. Watershed delineation was developed using as GIS platform to determine the watersheds' drainage areas. Quantifying this information will support the identification of which of the three watersheds contribute the most to water impairments in the Lower Laguna Madre by assessing each watershed independently. The watershed characterization has been shown to support stakeholders in the region for optimal watershed management and enhance their decision-making process.

## 2. Study Area

The Laguna Madre is composed of two sections: The Upper Laguna Madre and the Lower Laguna Madre (LLM). The Laguna Madre is also unusual for being one of only five hypersaline coastal ecosystems in the world [31,32]. This estuary encompasses 20% of Texas's protected coastal waters while contributing 40–51% of the state's commercial fish catch historically as well as providing a common ground for migratory birds [1,32,33]. The LLM is the area of interest in this study since the north and central watersheds inflow to two of the three segments that are currently considered impaired. The north and central watersheds encompass an area of 3116 km$^2$ located in South Texas in the northern and central area of the LRGV region. The LRGV is a semiarid region in South Texas bordered by Mexico to the south and the Gulf of Mexico to the east [16]. This watershed is comprised of three main waterways: HWMD in the southwest extending to the east, RVD in the north, and IBWCNF in the southeast (Figure 1). The study area takes up a large plain of South Laguna Madre Watershed Hydrologic Unit Code 12110208 (8-digit HUC). North and central watersheds encompass 37% of the area in the LLM watershed. The study area has significant hydrology challenges due to flat terrain, where previous studies will be considered when processing the data. Its elevation gradually slopes from 102 to 0 m with a high range of precipitation between 50–70 cm/year. The Arroyo Colorado is located south of the IBWCNF waterway. Although relatively close to one another, they are not considered intersecting. In general, soils in the LRGV region consist of calcareous to neutral clays, clay loams, and sandy loams [20]. Therefore, the low permeability of the soils influences the drainage characteristics.

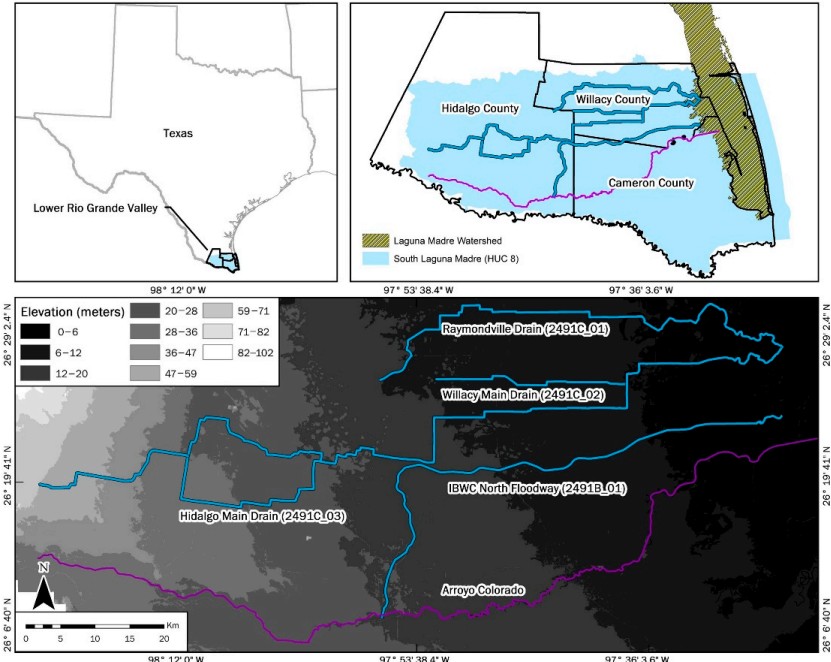

**Figure 1.** Location and elevation map of Study area is the North and Central watersheds located in the Lower Rio Grande Valley, of South Texas, Hidalgo/Willacy Main Drain (HWMD), Raymondville Drain (RVD), and International Boundary & Water Commission North Floodway (IBWCNF).

## 3. Methodology

The methodology to collect and analyze data for the characterization of the three watersheds was the acquisition of geospatial data and non-geospatial data. Geospatial data were obtained to develop a GIS database through a cyberinfrastructure to recognize the dominated attributes contributing to the watersheds. Therefore, the elaboration of watershed maps facilitated the identification of these attributes. Due to the wide inventory of data, a cyberinfrastructure was used to make data collection more efficient. Then, the elevation data were reconditioned to better represent the drainage areas of the watershed with respect to the terrain of the study area. In addition, NPS and PS pollution data were obtained to fully characterize the watersheds and to determine relative sources of pollution. Non-geospatial data were divided into two sections: water quality and flow data. Water quality was incorporated to determine the relationships between potential sources of pollution with the parameters found in each watershed. Available flow data were used to determine the load concentrations for each water quality parameter.

### 3.1. Cyberinfrastructure Development

In this study, cyberinfrastructure was established by developing the River and Estuary Observatory Network (REON) (http://dev.reon.cc:8607/ accessed on 17 August 2021). REON provides an extensive overview of all the available data from national, state, and local sources on this site. This platform helped in obtaining quality data for an overview of the north and central watersheds' characteristics, where stakeholders from the study area could support the characterization. The website now serves as a cyber-collaboratory platform for engaging stakeholders with an interest in data and information for a certain location [6]. Due to the wide inventory of data, the cyberinfrastructure also supported the acquisition of geospatial data, making the process more efficient which consisted of having all the geospatial data in only one source, REON. The value of the REON website in this study is that it portrays special features such as metadata, properties of the layers, and layer attributes to enhance watershed characteristics. The REON website was used to incorporate geospatial data and layers to show relative characteristics of the watersheds based on the watershed boundaries. To fully demonstrate watershed characteristics, the delineation of watershed boundaries was crucial for the assessment. Watershed delineation played an important role in this study, especially for the REON website to understand the extent of the study area.

### 3.2. Development of Watershed Delineation

The watershed delineation process is fundamental for the overall characterization to define the watershed boundaries and subwatersheds within each watershed. Generally, the watershed slopes from west to east through the heart of the LRGV, with an average slope of fewer than 0.3 m per kilometer [34]. Overall, its flat terrain varies from 0 m to 100 m. The resolution of the elevation raster-files was changed from 1 m to 60 m, which contributed to the reduction of file size and thus provided an efficient analysis. Since watershed delineation is key for this study, an ample watershed delineation was implemented to better assess the drainage areas of the watersheds. Previous studies have shown positive results for DEM reconditioning in watershed delineations in flat terrains [14]. Moreover, the assessment of satellite data and National Hydrography Dataset (NHD) was considered when evaluating the waterways and other laterals for the process. The satellite data were used to determine the accuracy of the location of the North and Central waterways. The NHD flowlines were used to determine the addition of laterals that could potentially drain into the waterways. LIDAR elevation data were reconditioned by developing several raster-elevation files to incorporate waterways into the data. This processing refers to burning waterways because the elevation data are not able to detect the waterways (Figure 2). Burning waterways consist of a rasterized version of the digital vector file to decrease the relative elevations of stream pixels by a uniform depth. Therefore, burning new channels

into the DEM is an attempt to force alignment between topographically derived flowlines and independently mapped hydrography [35].

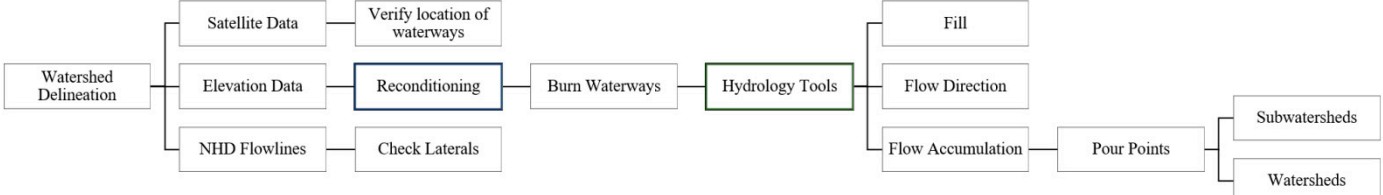

**Figure 2.** Watershed delineation methodology.

Once processing the LIDAR elevation data, the hydrology tools were used to develop elevation raster files such as fill, flow direction, and flow accumulation. Only three pour points were added manually to each corresponding waterway and then automated subwatersheds were developed. With the subwatersheds delineated, the overall watershed boundaries for the three watersheds were determined based on the flow accumulation lines. The flow accumulation lines correspond to the flow path for each watershed based on elevation data. The flow accumulation lines embody the actual waterways in mostly all the watersheds. The watershed boundaries correspond to the flowlines and follow an enhanced methodology for the type of terrain in the region.

### 3.3. Data Collection

The study was developed based on the guidelines of the United States Environmental Protection Agency (USEPA) Handbook for Developing Watershed Plans to Restore Our Waters [36]. A summary of the data used in the study can be found in Table 1. NPS pollutant loads through sediment and runoff courses are highly related not only to land use/cover characteristics but also to topography [37–39]. This study integrates land cover data from the 2016 National Land Cover Database (NLCD) [40] with a spatial resolution of 30 m to determine relative contributions of NPS pollution in the north and central watersheds. The land cover type data identified as NPS pollution encompass urban and agricultural areas only. Each watershed was treated individually to characterize the type of land cover in the area. The NPS pollutants identified within the watersheds were cultivated crops areas and urbanized areas and South Texas large ranches (STLR), species, wildlife management areas (WMA), Onsite Sewage Facility (OSSF), and colonias.

**Table 1.** Data sources used for characterization the IBWCNF, HWMD and RVD.

| Data | Source | Year | Usage |
|---|---|---|---|
| LIDAR Data | USGS, TNRIS | 2018 | Watershed Delineation |
| Hydrograph (NHD) | USGS | 2012–2019 | Watershed Delineation |
| Land Cover | NLCD | 2016 | NPS |
| STLR | TCEQ | 2018 | NPS |
| TLAP | TCEQ | N/A | PS |
| WWO | TCEQ | N/A | PS |
| MSW | TCEQ | N/A | PS |
| OSSF | Colonias | 2021 | NPS |
| MS4s | TCEQ | N/A | PS |
| Colonias | TCEQ | 2015 | NPS; OSSF points |
| Desalination Plants | TWDB | 2021 | PS |
| Address Points | TNRIS | 2018 | OSSF points |
| IBWC Gage Stations | IBWC | 2012–2020 | Flow data (IBWCNF) |
| SWQM Station | TCEQ | 2011–2019 | Flow and water quality (IBWCNF) |
| SWQM Stations | TCEQ | 2017–2019 | Flow and Water quality (HWMD and RVD) |

Cultivated crops and urban areas are two types of land cover that can be contributing to NPS pollution. Agricultural and stormwater runoff generated from cultivated crops and urban areas; respectively. Runoff carries various pollutants such as nutrients, sediments, heavy metals, and bacteria which have a negative impact on the receiving waterbodies [41]. In peri-urban areas, agricultural/rural NPS pollution and urban NPS pollution are two types of sources that have gained considerable concern because urban expansion and agriculture intensification may act as a source or sink for contaminants to move toward surface water bodies [42]. Agricultural and urban areas in a watershed have shown in previous studies to be the main contributors to NPS pollution. Another type of NPS pollutants source is the STLR. The main concern with this type of NPS pollutants is the exposure to several hazardous contaminants from the practice of livestock. The improper management of livestock wastes (manure) can cause surface and groundwater pollution [43]. Water pollution from animal production systems can be by direct discharge, runoff, and/or seepage of pollutants to surface or groundwater [44].

OSSFs are designed to treat domestic wastewater using a septic tank for screening and pretreatment and a drain field where pretreated septic effluent is distributed for soil infiltration and final treatment by naturally existing microorganisms [45]. Species with WMA were found close to the coast of each watershed. These NPS pollutants contribute to high bacteria loadings to waterbodies from wildlife in the region. Grazing animals and wildlife can also negatively affect the water quality of runoff and waterbodies with bacterial contamination [46]. In Texas, non-avian wildlife, such as deer or feral hogs, are commonly found to be significant contributors of bacteria to natural streams [43,46]. In addition, colonias are considered the most distressed areas in the United States. They are usually found along the U.S.–Mexico border, which often lacks necessities such as sewer systems, drinkable water, and overall sanitary housing. Many homes within colonias cannot meet county building codes because they lack indoor bathrooms and plumbing, a prerequisite for connection to local water lines and sewage systems [17]. Consequently, colonias can be a potential contributor of NPS pollutants since they lack adequate solid waste disposal and wastewater systems. TCEQ created a classification system to identify the colonias with adequate utilities and the ones that lack basic utilities. The red and yellow classification was the one selected for colonias that potentially carry NPS pollution. Based on the priority classification by the Rural Community Assistance Partnership, OSSFs located in the colonias having a health hazard (red colonias) were assumed to have a greater failure rate of 70%. Conversely, a 30% failure rate (determined based on local expert knowledge) was assigned to areas having the lower priority ratings (yellow colonias) [47]. The term "colonia" refers to a settlement or neighborhood that is an unincorporated rural and peri-urban subdivision along Texas' border with Mexico [48].

STLR and colonias were extracted from TCEQ NPS Pollution database. There are currently limited studies in quantifying NPS pollution in semi-urban areas such as LRGV, where the topography is relatively flat. Furthermore, species and wildlife management areas WMA were considered as well as part of the NPS pollution for the effort in assessing their contaminants to the waterbodies. These were extracted from Texas Parks and Wildlife Department (TPWD). In addition, OSSF locations were mainly extracted from the colonias layer that identified OSSF as their wastewater collection facility. In Jeong's study [47], they utilized a methodology to extract OSSFs from merging address points with colonias. To estimate the number of OSSFs within the watershed, 911 address data for Cameron, Willacy, and Hidalgo counties were obtained. The address points represent the number of homes within a specific area. Combing this layer with the colonias areas, the acquisition of OSSFs was achieved. The colonias layer provided information about this classification and identified the type of colonias with limited wastewater disposal as well as adequate solid waste disposal. OSSFs were extracted from the red and yellow classification from colonias as well as the wastewater community section for onsite systems.

With the collaboration of local stakeholders and state-wide resources, the compilation of PS pollutants was obtained. The PS of pollutants identified in the north and central

watersheds include permitted wastewater outfalls (WWO), Texas Land Application Permit (TLAP), Municipal Solid Waste (MSW), Municipal Separate Storm Sewer System (MS4), and desalination plants [48].

There is a substantial contribution of bacteria from wastewater outfalls, which potentially discharges to the waterways. Fecal contamination of water normally results from direct entry of wastewater from a municipal treatment plant into a water body [46,47]. There were two types of WWOs identified in these watersheds: domestic and industrial wastewater discharge. Domestic WWOs discharge less than 1 million gallons per day (MGD) while the ones with a discharge greater than 1 MGD may be either domestic sources or industrial wastewater treatment plant effluent. According to TCEQ, TLAP refers to the spreading of sewage from several applications, such as surface irrigation, evaporation, drain fields, or subsurface land application [49]. MSW facilities not only affect the surface water within the watershed but also groundwater. Closed landfills are commonly unlined and poorly capped and may be sources of a large number of organic compounds to surrounding groundwater and surface water [50]. Polluted stormwater runoff is commonly transported through MS4s and then often discharged, untreated, into local water bodies [51]. MS4s are identified to discharge significant levels of contaminants to waterbodies in the United States and are now one of the major sources of water pollution in the nation [24]. Information about desalination plants was obtained from the Texas Water Development Board (TWDB) to support the PS pollution contribution to the watersheds. Disposing the concentrate from the desalination plant in the surface water is the most common method of concentrate disposal which is considered a point source [52]. These sources can be potential contributors to water quality impairments to the North and Central waterways.

Water quality data were obtained for the three watersheds from the Surface Water Quality Monitoring Information System (SWQMIS) database. The TCEQ maintains SWQMIS database to serve as a repository for surface water data throughout Texas. All the data available in the SWQMIS database have to be collected according to TCEQ surface water quality monitoring standards. Moreover, data must be verified and validated prior to its loading into SWQMIS. HWMD has a TCEQ monitoring station (ID 22003) located at FM 1420 1.65 KM south of the intersection with FM 490 east of Raymondville (Figure 3). In addition, RVD has a TCEQ monitoring station (ID 22004) located at Willacy County Road 445 800 m north of the intersection with FM 3142. Both HWMD and RVD monitoring stations have 8 water quality samples available on the SWQMIS database. Data from both sites were collected by Clean River Programs (CRP) from 2017 to 2019 [53]. For IBWCNF, one TCEQ monitoring station was installed to collect water quality data since 2012. IBWCNF station ID is 20930 and is located at US 77 2.5 KM south of the intersection of US 77 and FM 2629 in the city of Sebastian. There were 25 water quality samples for the IBWCNF watershed available from SWQMIS from 2012 to 2019 [54,55]. The water quality parameters assessed in this study include the following: bacteria, ammonia, total Kjeldahl nitrogen (TKN), total phosphorus (TP), chlorophyll-a, nitrite, and nitrate. On the other hand, there is currently limited flow data for HWMD and RVD waterways since the monitoring in both stations started in 2017. The data were quantified on a quarterly basis for the period of two years. However, IBWCNF has a flow monitoring station (ID 08470200) installed by USIBWC at the same location of the SWQM near Sebastian that collected data from 2012 to 2020.

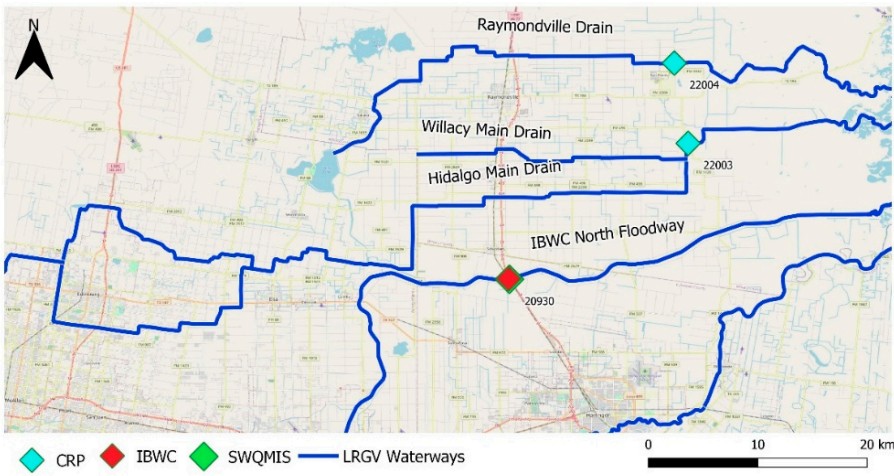

**Figure 3.** Location of water quality and flow data monitoring stations.

## 4. Results

### 4.1. REON Cyberinfrastructure

With the collaboration of REON, a cyberinfrastructure website, both data collection and the development of maps were accomplished. This platform provided an efficient watershed characterization by exposing significant guidelines from the EPA watershed characterization manual. This manual provides the basis to meet water quality and watershed management goals. Physical and natural features, land use, waterbody conditions, pollutant sources, and waterbody monitoring information are the data needed to characterize a watershed [56]. The first step for the watershed characterization was to develop the watershed delineation for the three watersheds. The results were then uploaded to the REON website to show watershed boundaries. Additionally, NPS and PS pollution layers were included in each watershed to facilitate the characterization process based on EPA watershed characterization. The cyberinfrastructure gathers existing watershed boundaries, hydrology, land use, NPS pollution, PS pollution, water quality stations, and flow stations to support the overview of the watershed characteristics. Three maps were created: Watershed delineation results, NPS pollution, and PS pollution maps. The maps created facilitated the watershed characterization by integrating geospatial data for NPS and PS pollutants for each watershed individually. The development of maps portrayed in the cyberinfrastructure helped stakeholders collaborate in the characterization by providing inputs for each potential source that could contaminate the area. The web user interface at the regional level is available for every stakeholder regardless of time or location.

### 4.2. Watershed Delineation

This section introduces the watershed delineation results for the study area (Figure 4) (Table 2). The watershed delineation encompassed a comprehensive LIDAR elevation data reconditioning to well display the North and Central Watersheds' characteristics. Elevation reconditioning has revealed improved results in areas with very flat terrain. Previous studies had positive results with respect to their watershed delineation by performing this methodology [13]. Burning the waterways to the elevation data has enhanced the terrain to better support the current conditions of the elevation changes in the waterways. Generally, all the waterways within the area are man-made, which makes it challenging for the elevation data to capture the waterways. The north and central watersheds presented a total area of 3116 km$^2$ of which HWMD watershed presented an area of 1357 km$^2$, RVD watershed is 1021 km$^2$, and IBWCNF watershed is 737 km$^2$ (Table 2). HWMD watershed covers 68% of its area in Hidalgo County, 31% in Willacy County, and 1% in Cameron County. This watershed covers a wide central area of the LRGV region. It extends across nine cities in the region. Moreover, it covers the McAllen-Edinburg-Mission Metropolitan Statistical Area (MSA) of the LRGV region, which is ranked the 5th largest in the state of

Texas. The RVD watershed, located in the north area of the LRGV region, covers 30.7% in Hidalgo County, 68.9% in Willacy County, and 0.4% in Kennedy County. The city of Raymondville, San Perlita, and a northeast portion of the city of Edinburg are the only cities within the watershed.

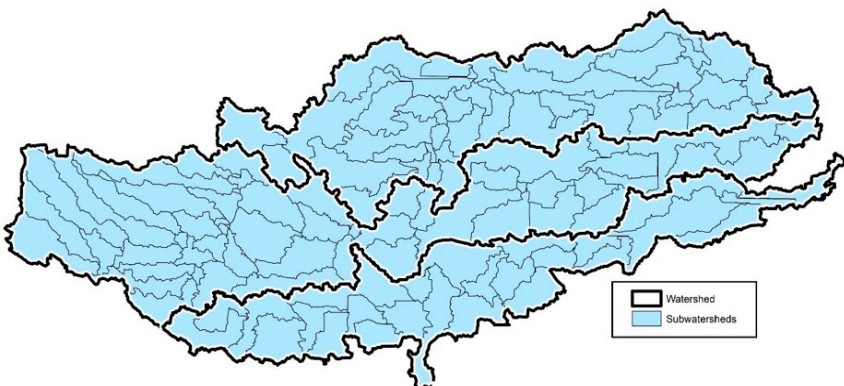

**Figure 4.** North and Central subwatersheds within each of the watersheds; top: Raymondville Drain (RVD); middle: Hidalgo/Willacy Main Drain (HWMD); bottom: International Boundary & Water Commission North Floodway (IBWCNF).

**Table 2.** Watershed delineation results for the three waterways.

|  | **HWMD** | **RVD** | **IBWCNF** |
|---|---|---|---|
| Watershed Area (km$^2$) | 1357 | 1021 | 737 |
| Number of Sub-watersheds | 91 | 72 | 73 |
| Hidalgo County | 68% | 31% | 52% |
| Willacy County | 31% | 69% | 24% |
| Cameron County | 1% | 0% | 24% |

IBWCNF watershed is located 53% in Hidalgo County, 24% in Willacy County, and 24% in Cameron County. This watershed is within the southern area of the North and Central Watersheds and intersects with the Arroyo Colorado Watershed. Eight cities are included in the IBWCNF watershed. The IBWCNF branches off of the Main Floodway at the Llano Grande, a shallow lake located southwest of the city of Mercedes [57]. The IBWCNF Waterway is considered a man-made waterway approximately 77 km long and is used to divert the Arroyo Colorado's flow. The city of Mercedes is upstream of IBWCNF flow and downstream of the Arroyo Colorado Waterway when the flow is exceeding its capacity. During flood conditions, which the IBWC defines as flow exceeding 40 cubic meters per second, approximately 80% of the flow in the Arroyo Colorado is diverted to the IBWCNF [58].

*4.3. Nonpoint Sources*

In this section, the watershed sources that potentially contribute the most to NPS pollutants were identified. Table 3 shows the results of the ratio of PS and NPS pollution sources to the area of each watershed. The predominant land cover for the North and Central Watersheds is cultivated crops representing 53% of the total area located mostly in the northeast sector of the watersheds. This type of land use is within the downstream tributary areas of the watersheds. Urbanization areas within the North and Central Watersheds cover 13% of the total area. STLR were found near the coast of the three watersheds.

**Table 3.** Ratio of NPS and PS pollution sources with respect to the area of each watershed.

|  | Sources | HWMD | RVD | IBWNF |
|---|---|---|---|---|
| Nonpoint Source Pollution | Urbanized Areas | 0.20 | 0.05 | 0.24 |
|  | Cultivated Crops | 0.47 | 0.52 | 0.59 |
|  | STLR | 0.06 | 0.20 | 0.04 |
|  | Species * | 0.03 | 0.10 | 0.20 |
|  | OSSFs | 3.38 | 0.05 | 6.13 |
|  | Colonias | 0.25 | 0.01 | 0.29 |
| Point Source Pollution | Texas Land Application Permit | 0.006 | 0.004 | 0.004 |
|  | Wastewater Outfalls | 0.008 | 0.005 | 0.012 |
|  | Municipal Solid Waste | 0.013 | 0.004 | 0.004 |
|  | MS4 Permit | 0.006 | 0.001 | 0.016 |
|  | Desalination Plants | 0.001 | 0.001 | 0.003 |

* Quantified data.

About 73% of the HWMD watershed area is covered with NPS pollutants sources. The watershed's cultivated crops correspond to approximately 47%, and 20% of urbanized areas. Urban growth in the watershed will primarily occur in areas that are currently cultivated and will influence the region's water quality [34]. Therefore, the HWMD watershed was identified with the highest ratio of urban areas among the other watersheds, with respect to their watershed area. The watershed encompasses 6.4% of STLR areas. Only El Suaz ranch pertains to the watershed. These STLR areas have grazing livestock activities that ultimately carry significant levels of bacteria. There were 46 species identified in this watershed along with two WMA units. La Palomas units, Longoria, and Fredrick, were found to possess hunting activities for their diversity of species. A total of 4591 OSSFs were found in the HWMD watershed from a total of 9170 in the north and central watersheds. All OSSFs have a potential for adverse environmental impact if they are improperly functioning, but those closer to streams present an elevated risk [34]. The watershed has 336 colonias, where 80 are classified with limited solid waste disposal, and 33 lack adequate solid waste and wastewater disposal. The total area of the colonias in the watershed is 26.8 km$^2$.

NPS pollutants sources cover almost 86% of the total area of the RVD watershed. The watershed has 51% of cultivated crops and only 2% of urban areas. The RVD watershed encompasses 19% of STLR areas. King Ranch, East Foundation, and El Suaz are the ranches that cover the watershed. Not only agriculture activities take place within the STLR areas. Livestock also grazes in this area, which can increase the relative contribution of bacteria. Fecal pollution brought to the rivers through surface runoff and soil leaching represents the NPS pollution; its origin can be the wild animals and grazing livestock feces and cattle manure spread on cultivated areas [50–52]. A total of 56 OSSFs were identified in the watershed. The RVD watershed has only 13 colonias recorded from which 1 is limited to solid waste disposal and 3 lack of basic utilities. Colonias within the watershed cover an area of 21.6 km$^2$.

The IBWCNF watershed corresponds to 73% of cultivated crops and 13% of urban areas. This watershed has the highest ratio of agricultural lands that can be a possible source of ammonia and nitrogen in the surface water. According to the EPA, watersheds could be affected by the level of decomposition of organic matters and some fertilizers used in agriculture. This watershed covers a portion of El Suaz ranch with 5% of STLR areas. There were 4523 OSSFs identified in this watershed, corresponding to a 6.33 ratio between the total OSSFs and the total area of the watersheds. The colonias cover an area of 23.4 km$^2$ within the IBWCNF watershed. This watershed has 216 colonias from which 65 lack proper solid waste disposal, and 51 lack both solid waste and wastewater disposal.

In summary, the HWMD watershed was identified with the highest ratio of urban areas among the other watersheds with respect to their watershed area. The identification of McAllen-Edinburg-Mission MSA in this watershed demonstrates the high presence of urban areas. The HWMD had 20.3% of urban areas and 8.8% from the three watersheds.

In contrast, the IBWCNF presented a higher percentage of 24.3% in urban areas, but it only had 5.8% with respect to the overall area of the North and Central Watersheds. The RVD and IBWCNF watersheds were the ones to have greater NPS pollution derived from cultivated crops [46]. The RVD watershed was the highest with STLR areas.

### 4.4. Point Source

The HWMD watershed has a total of 11 WWOs from which 5 were found to discharge less than 1 MGD, and the rest discharged more than 1 MGD. Major PS pollutants identified in this watershed were TLAP and MSW. The TLAP corresponds to the presence of high levels of nitrogen in the watershed, and the MSW corresponds to the presence of high total phosphorus levels. There were 8 TLAPs found upstream of the watershed. Currently, there are 2 active MSW facilities in the HWMD watershed. This watershed has a total of 17 MSW facilities recorded from which 4 are considered closed facilities, 4 are inactive, 2 posted closed, and the rest are not constructed. HWMD watershed covers 13% of MS4s. There are currently 7 MS4s permitted areas within the HWMD watershed. The HWMD watershed has the highest MS4s areas among the other watersheds. Therefore, the HWMD watershed shows severe impact by the PS pollution compared to the other watersheds

Although the RVD watershed has a greater area compared to the IBWCNF watershed, it is limited with PS pollution (Figure 5). Five WWOs were identified within the watershed boundaries from which 3 are considered industrial wastewater effluent and 2 domestic. Only 4 TLAPs were found in the RVD watershed. Currently, the City of Edinburg Landfill is an active MSW in the RVD watershed. A total of 4 MSWs were identified in the RVD watershed: 2 not constructed, 1 closed, and 1 post closed MSWs. RVD watershed is considered to contribute to 0% of MS4s, with only 0.3% of the city of Edinburg's MS4 found. This watershed covers almost the entire Willacy County, which is identified as limited in MS4s. The IBWCNF watershed presents 9 WWOs from which 4 are domestic and 5 are industrial wastewater effluent. For instance, only 3 TLAP were found, and 3 active MSWs were identified. These PS pollutants are mainly located upstream of the watershed. As a result, it is important to identify the potential PS pollutants of the downstream area of the Arroyo Colorado Watershed that diverts to the IBWCNF watershed. The IBWCNF watershed has 7% of MS4s permitted areas. The MS4s permitted areas include 11 cities. Consequently, it is important to improve stormwater management within these areas to mitigate PS pollutants. Unlike sanitary sewer systems, MS4 systems do not treat the stormwater collected; instead MS4s are required to develop and implement stormwater management programs (SWMP) that reduce the amount of contaminants that enter the system and prohibit illicit discharges [24].

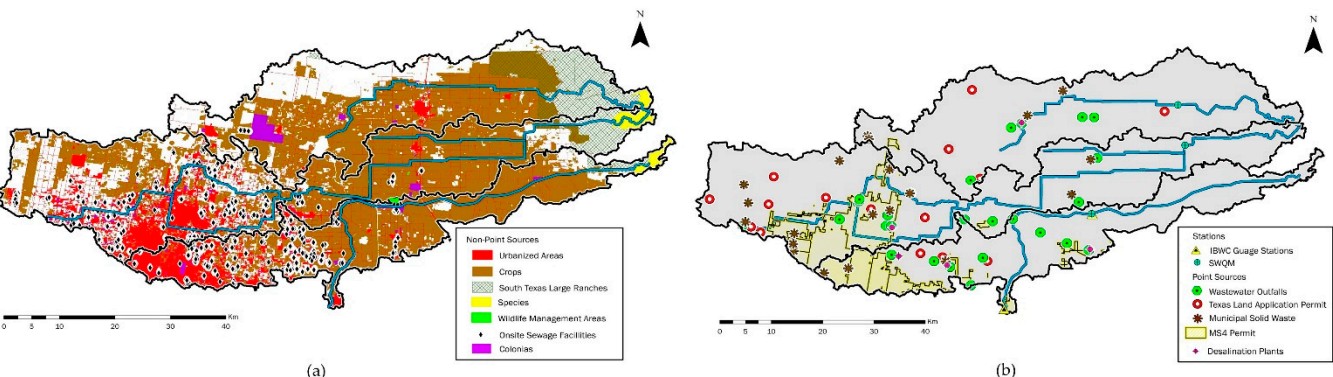

**Figure 5.** Location of potential sources of pollution in the North and Central Watersheds: (**a**) non-point sources and (**b**) point sources of pollution identified in each watershed.

### 4.5. Water Quality Parameters

The water quality parameters samples for the north and central watersheds are shown in Figure 6, where the red line represents the screening level according to TCEQ water quality standards. HWMD watershed has *E. coli* levels higher than the screening level of 126 MPN/100 mL from 2017 and 2019 [2]. In 2019, the *E. coli* levels were above 2000 MPN/100 mL. The existence of high levels of bacteria is caused by a variety of NPS and PS pollution sources such as urban runoff, agricultural lands, ranches, WWO, OSSF, MS4s, and colonias. Ammonia levels in this watershed were below the screening level with 2.7 mg/L as N, which is considered the highest record. In 2018, the TKN levels were the highest compared to the other years with more than 3.0 MGL as N. The presence of TKN in the HWMD watershed, according to the EPA, can be traced to failing septic systems, croplands, and industrial discharges [59]. TP levels barely exceed the screening level of 0.7 mg/L with the maximum value of 0.8 mg/L in 2017. Moreover, the nitrite and nitrate levels found in the watershed are higher than the screening level of 1.95 mg/L [2,60]. Chlorophyll-a levels identified surpassed the screening level of 14 μg/L for the three years [2]. In 2018, chlorophyll-a had the highest level of 98 μg/L.

The RVD watershed had higher levels of *E. coli* compared to the other watersheds, which suggests that there could be several sources of NPS and PS such as septic tanks that can be leaking. Further, sewage may overflow from poorly structured sewage systems and create polluted stormwater runoff [61]. However, ammonia levels for the RVD watershed are acceptable since they are below the screening level of 0.33 mg/L with a maximum value of 0.2 mg/L in 2018 and 2019 [60]. The TKN levels mainly surpassed the screening level of 1.0 mg/L in 2018 and 2019. TP levels were lower in all the years recorded, with a maximum value of 0.4 mg/L in 2019. According to the USGS report, bank erosion is the main source of total phosphorus during flooding events that can be the potential source in these watersheds [62]. Nitrite and nitrate levels surpassed only in 2017, but the highest level identified was almost 6 mg/L as N in 2019. For Chlorophyll-a levels, the RVD watershed showed its highest level of 70 μg/L in 2019.

In the IBWCNF watershed, the levels of bacteria were identified to be higher in 2013, 2014, 2015, and 2019. The highest level was around 8000 MPN/100 mL in 2013. The bacteria levels from 2016 through 2018 were determined to be slightly below the screening level of 126 MPN/100 mL. The results showed, according to Olmstead [46], that the watershed is affected by wildlife with small contributions of domestic animals and point sources. The ammonia levels were identified to be less than the screening level during all the years. This finding indicates that the watershed is limited to carrying significant levels of ammonia from agricultural runoff. TKN levels have shown to be relatively higher than the screening level with the highest of 2 mg/L as N in 2018. High levels of total nitrogen are caused by the decomposition of detritus and any anthropogenic loadings [63]. High levels of total nitrogen are caused by the decomposition of detritus and any anthropogenic loadings [63]. TP levels were lower than the screening level of 0.7. The IBWNF watershed is limited to algae growth since TP levels are low. Nitrite and nitrate levels are higher than the screening levels; 7 mg/L was the highest level recorded in 2015. Chlorophyll-a levels were determined to be higher than the screening levels for nearly all the years. This finding indicates the presence of excess quantities of algae [64].

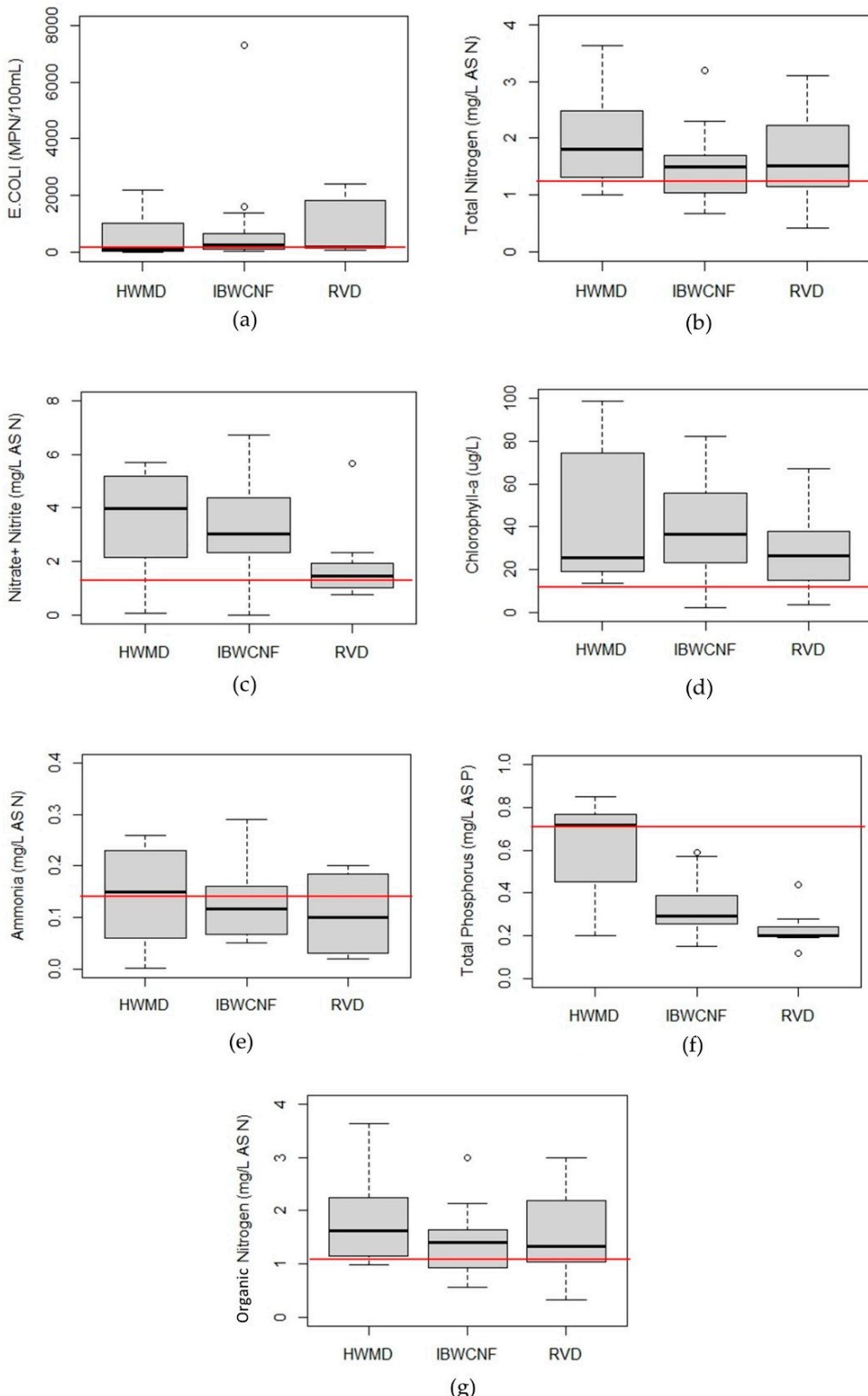

**Figure 6.** Comparison between water quality concentration levels for Hidalgo/Willacy Main Drain (HWMD), Raymondville Drain (RVD), and International Boundary & Water Commission North Floodway (IBWCNF)): (**a**) *E.coli*, (**b**) total nitrogen, (**c**) nitrate + nitrite, (**d**) chlorophyll-a, (**e**) ammonia, (**f**) total phosphorus, and (**g**) organic nitrogen. Redline represents the TCEQ screening level for each parameter (data source: SWQMIS).

### 4.6. Flow Data

　　Waterbody monitoring data are used to portray historical data that would represent most conditions of the study area. Flow data encompassed the volumetric flow rate for each waterway recorded from each station available. HWMD waterway flow data reflect high flow values in 2019 with a mean value of 12 CMS, and in 2018 the mean value was below 10 CMS. These levels reflect a high correlation with flooding patterns with respect to sudden storm events from those years. Moreover, the RVD flow data showed high flow values in 2018 of almost 10 CMS (Figure A1 in Appendix A). Both HWMD and RVD flow data correspond to past abnormal flooding events in the LRGV region. The region has experienced high storm events since 2018 with over 38.1 cm to 50.8 cm of rainfall causing severe flooding damage [65]. Such flooding's caused a halt to everyday functions for weeks and months because of minor to destructive varying degrees of flood damage in city roads, frontage roads, residences and businesses, and infrastructure in the LRGV region. Hidalgo, Cameron, and Willacy counties have received the Presidential Disaster Declaration in which have been determined to be the most impacted areas [66]. There are limited data for this watershed since they are only available for three years with limited monitoring campaigns. Therefore, among the three watersheds, it has been determined that the HWMD waterway has the highest flow values that affect the loadings even if the water quality concentrations are low.

　　The IBWCNF watershed has two stations: Mercedes and Sebastian. However, only the flow values utilized for further analysis were the ones from Sebastian since the water quality samples were obtained near that station. This finding would represent a better overview of the IBWCNF watershed behavior with respect to load concentrations. In 2017 and 2018, flow data measured were more than 10 CMS. The flow values throughout 2012 to 2020 seem to have mean values below 5 CMS, which suggests a constant uniform flow for this watershed.

### 4.7. Pollutant Loadings

　　Pollutant loading calculations were obtained from quantifying flow and water quality data. To well represent the loadings with each respected watershed, the pollutant loadings were based on the watershed area for the three watersheds. Table 4 shows the results for the unit area loading rates for each watershed reflecting which of the three watersheds has the highest loading. The HWMD watershed shows higher results with respect to the flow, water quality parameters, and the overall watershed area, where both NPS and PS pollution are potential attributes of these elevated results. These data are not representative of the whole profile of the watersheds. More data should be quantified to better distinguish which watershed contributes the most to water impairments to the LLM.

**Table 4.** Summary of the pollutant loading ($kg/km^2/year$) for the three watersheds.

| Water Quality Parameters | HWMD | RVD | IBWCNF |
|---|---|---|---|
| Bacteria (Log *E.coli*) [1] | 12.8 | 12.3 | 12.4 |
| Ammonia | 121 | 31 | 48 |
| TKN | 1586 | 670 | 477 |
| Organic Nitrogen | 1466 | 639 | 429.4 |
| TP | 519 | 63.3 | 122.6 |
| Nitrite + Nitrate | 2950 | 581.46 | 1512.10 |
| Chlorophyll-a | 32.6 | 9.9 | 13.2 |

[1] Bacteria loading unit is in $MPN/km^2/year$. Source: SWQMIS.

　　The pollutant loadings per unit area distribution for each water quality parameter were provided with respect to each watershed area (Figure 7). Methods for calculating the loadings for each pollutant can be found in the USEPA Handbook for Developing Watershed Plans to Restore Our Waters [36]. These loadings were generated automatically through ArcGIS properties to show the difference among pollutant loadings. Bacteria load-

ings per unit area were determined to be slightly higher for the IBWCNF watershed than RVD. However, IBWCNF has more potential NPS and PS sources for bacteria than RVD. The mean value for the bacteria loadings in IBWCNF and RVD was 12.4 (kg/km$^2$/year) and 12.3 (kg/km$^2$/year); respectively. This can be explained by the fact that the main bacterial sources in both watersheds come from agricultural activities. The ratio in cultivated crops in IBWCNF was slightly higher than RVD. IBWCNF covered 59% of cultivated crops, while RVD covered 52%. Additionally, the flow volume in RVD was higher than IBWCNF. The average flow rate in RVD was 2.57 CMS, while in IBWCNF it was 2.38 CMS. This could be the reason why the bacteria loadings in both watersheds have a minor difference. TKN results proved to be higher for the HWMD, which support the relative contribution of the TLAP to this watershed. Nitrate and nitrite and chlorophyll-a concentrations were high in the HWMD, corresponding to the significant presence of urban areas in the watershed. Ammonia results showed to be higher in the IBWCNF watershed, supporting the identification of a substantial percentage of agricultural lands. The HWMD had the highest loadings for TP and organic nitrogen, supporting the presence of MSWs. Figure 6a–g reflects the loading with respect to the subwatersheds of the three North and Central Watersheds. The HWMD watershed was identified to be higher in all the water quality parameters due to the high flow recordings in this watershed.

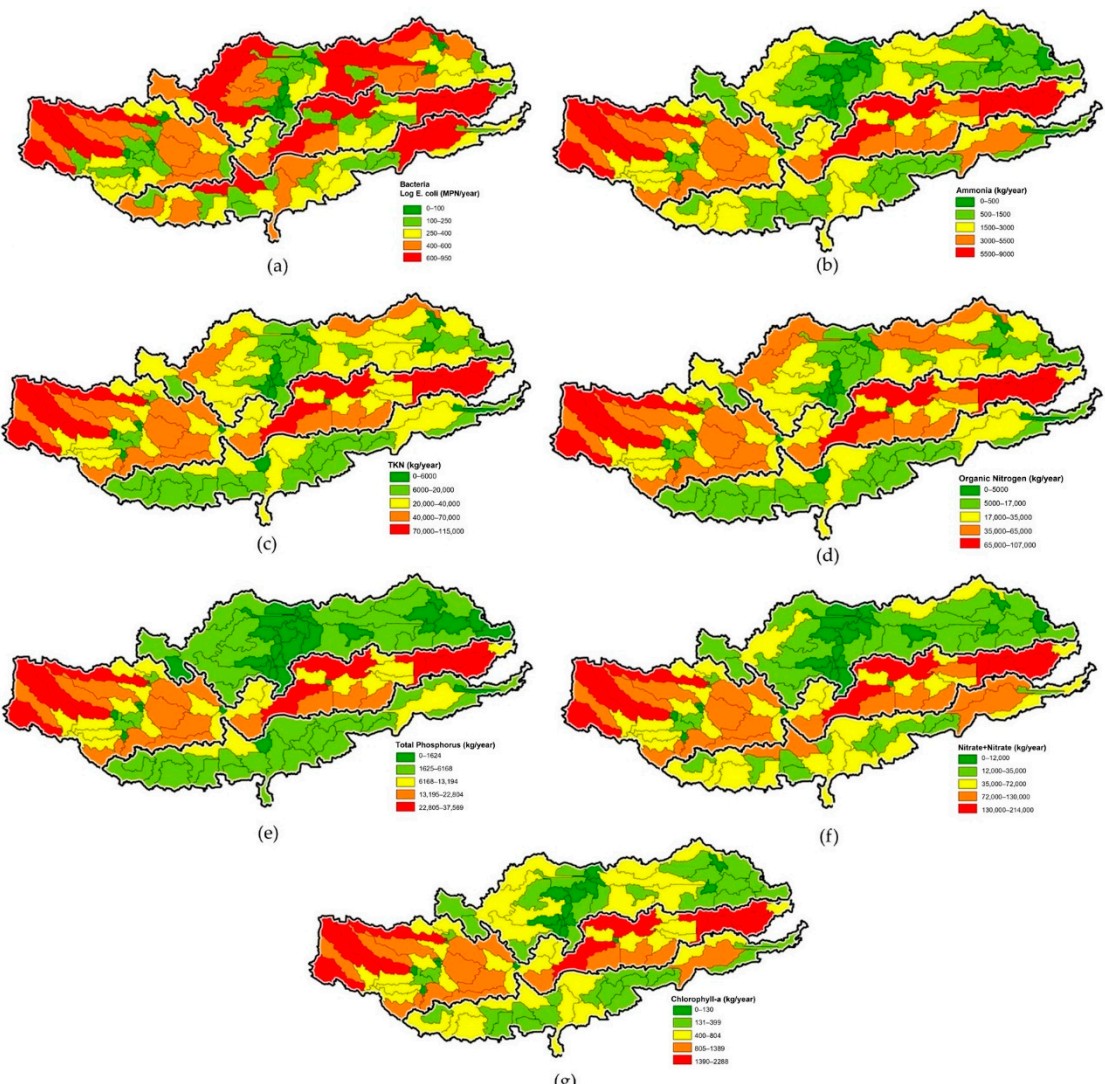

**Figure 7.** Spatial distribution of the pollutant loading for different parameters in the North and Central Watersheds: (**a**) bacteria, (**b**) ammonia, (**c**) total nitrogen, (**d**) organic nitrogen, (**e**) total phosphorus, (**f**) nitrite + nitrate, and (**g**) chlorophyll-a.

## 5. Discussion and Conclusions

The cyberinfrastructure and REON website contributed significantly to this study in portraying relevant characteristics of each of the North and Central Watersheds. The REON website not only collects distinct information into one single source but also allows the stakeholders within each watershed to assess the watershed characteristics. Therefore, this platform is an innovative tool that supports effective watershed characterization. ArcGIS automated hydrology tools have shown to have satisfactory results in delineating watersheds. Overall, the study showed that the watershed delineation process used provided acceptable results to characterize the North and Central Watersheds.

Although the HWMD watershed was not the highest regarding the urban areas, it is considered higher in NPS pollution with respect to the entire area of the North and Central Watersheds. Urban areas have more impact on the HWMD in comparison to the other watersheds regarding the overall watershed areas. This finding suggests that urban areas in this watershed are linked to the presence of bacteria and chlorophyll-a. Based on the water quality data obtained, only chlorophyll-a levels were higher than the other watershed levels. The high levels of chlorophyll-a relate to the HWMD watershed in extensive urban areas. Based on the total PS pollution found in the North and Central Watersheds, HWMD is the watershed to contribute a 3.66 ratio with respect to the watershed area. While this watershed has greater PS pollution than the other two watersheds, it is not particularly the most affected watershed with respect to the drainage area. The NPS and PS results for HWMD were consistent with the elevated levels of the water quality data analyzed from the SWQMIS database. Bacteria, total nitrogen, nitrate and nitrite, chlorophyll-a, ammonia, total phosphorus, and organic nitrogen in HWMD had significant values in this watershed compared to the other watersheds. In addition, the high pollutant loadings in this watershed correspond to the high flow values recorded. Therefore, more flow data are needed in the future to further support this characterization and make the proper connections between sources of pollution and pollutant loads.

The RVD watershed had a higher percent of 20.3% for ranches and was identified to be higher regarding the total area of the North and Central Watersheds as well. The water quality parameters associated with the presence of ranches are bacteria, ammonia, TP, nitrite, and nitrate. The results showed that the RVD watershed has greater bacteria levels in comparison to the other watersheds, which suggests ranches and the activities within these areas are causing high levels of bacteria. The RVD watershed pollutant loadings were generally low, but bacteria loadings were significant because of the high presence of NPS pollutants. Bacteria loading mean value corresponds to almost 12.3 MPN/km$^2$/year.

The IBWCNF watershed was identified to have higher crop areas with 58.5% regarding the area as well as the overall area of the three watersheds, which suggests the presence of significant agricultural activities. Therefore, it was determined that agricultural runoff is prone to release higher levels of ammonia where this watershed was limited to carry high ammonia levels. This finding indicates a possible change in land cover from 2016 to 2020. In addition to ammonia, bacteria, TKN, TP, nitrite and nitrate, and chlorophyll-a are present in agricultural areas. The IBWCNF watershed has a greater presence of nutrient water impairments because of the high agricultural area. This finding suggests the high levels of nitrite and nitrate in this watershed correspond to agricultural lands. This watershed had the higher contribution of PS pollutants such as WWO, OSSFs, MS4s, and colonias among the watersheds. The sources contributing to the high levels of water quality concentrations were identified. Ammonia, nitrate, and nitrite primary sources can be related to WWO, MS4s, and colonias. The load concentration results showed the IBWCNF to have high bacteria and ammonia loads. This finding suggests that the presence of a significant contribution of OSSFs is linked to bacteria loadings.

To uncover which North and Central watersheds contributed the most to the LLM watershed impairment, a cyberinfrastructure was established along with an ample watershed delineation. Then NPS pollution, PS pollution, water quality concentrations, flow data, and pollutant loadings were enhanced to identify unique characteristics of the watershed.

HWMD and IBWCNF were the watersheds to contribute the most in water impairments to the LLM watershed. They were found to have significant loadings of water quality parameters as well as NPS and PS pollutant contributions. Urban areas, TLAP, and MSW were related to the high contribution of chlorophyll-a, TKN, and TP. OSSFs and colonias were linked to the major influence of bacteria concentrations and loadings of which the IBWCNF watershed possesses the most. These results along with the user-friendly cyberinfrastructure may assist stakeholders from the region in identifying the characteristics of watersheds and mitigate the sources of pollution. This study is essential in bringing awareness to the local communities that reside within these watersheds, especially the people who visit the LLM watershed. One of the limitations of this study was the acquisition of available data for such an extensive study area of more than 3000 km$^2$. Additional flow data and water quality data could enhance the characterization as it was limited to only 8 samples for the HWMD and the RVD watersheds. Flow data are essential for determining the load concentrations and provide a better overview of the north and central watersheds' potential sources of pollution.

**Author Contributions:** Conceptualization, A.E. and A.M.; methodology, A.M., A.O. and A.E.; software, A.E. and C.F.; formal analysis, A.O. and L.N.; writing—original draft preparation, L.N., J.J., I.R.S.C. and A.M.; visualization, I.R.S.C.; supervision, A.M. and A.E.; project administration, J.J. All authors have read and agreed to the published version of the manuscript.

**Funding:** Funding for this research was provided by the Texas Commission on Environmental Quality (Project Contract# 582-19-90196) and financed through grants from the U.S. Environmental Protection Agency (Federal ID# 99614623).

**Institutional Review Board Statement:** Not applicable.

**Informed Consent Statement:** Not applicable.

**Data Availability Statement:** The data presented in this study are available on request from the corresponding author.

**Acknowledgments:** The authors would like to thank Tim Cawthon (Texas Commission on Environmental Quality) for his support and contribution in data for the study. The authors also would like to thank Javier Guerrero and the Lower Rio Grande Valley Stormwater Taskforce members for their collaboration in the study.

**Conflicts of Interest:** The authors declare no conflict of interest.

## Appendix A

**Table A1.** Hidalgo Willacy Main Drain Water Quality.

| Date | Bacteria | Ammonia | TKN | TP | Nitrite | Nitrate | Chlorophyll-a |
|---|---|---|---|---|---|---|---|
| 10/4/2017 | 610 | 0.02 | 1 | 0.733 | 3.02 | 0 | 57 |
| 12/3/2017 | 10 | 0.26 | 2.85 | 0.847 | 3.87 | 0 | 13.5 |
| 5/1/2018 | 120 | 0.002 | 3.63 | 0.755 | 4.71 | 0 | 91.5 |
| 7/18/2018 | 20 | 0.2 | 2.1 | 0.2 | 1.2 | 0.099 | 98.5 |
| 10/31/2018 | 80 | 0.1 | 1.5 | 0.67 | 5.6 | 0.09 | 23.9 |
| 1/29/2019 | 31 | 0.1 | 1.21 | 0.7 | 5.6 | 0.06 | 19.3 |
| 4/2/2019 | 1400 | 0.2 | 1.4 | 0.78 | 4.02 | 0.06 | 27 |
| 7/16/2019 | 2200 | 0.26 | 2.1 | 0.23 | 0.03 | 0.02 | 19.3 |

**Table A2.** Raymondville Drain Water Quality.

| Date | Bacteria | Ammonia | TKN | TP | Nitrite | Nitrate | Chlorophyll-a |
|---|---|---|---|---|---|---|---|
| 10/4/2017 | 1940 | 0.02 | 1 | 0.28 | 1.17 | 0 | 36.3 |
| 12/3/2017 | 150 | 0.1 | 0.42 | 0.2 | 1.52 | 0 | 18 |
| 5/1/2018 | 220 | 0.02 | 2.75 | 0.12 | 2.34 | 0 | 33.3 |
| 7/18/2018 | 150 | 0.1 | 3.1 | 0.2 | 0.8 | 0.05 | 39.8 |
| 10/31/2018 | 1700 | 0.2 | 1.3 | 0.2 | 1.5 | 0.05 | 11.7 |
| 1/29/2019 | 74 | 0.17 | 1.43 | 0.2 | 5.6 | 0.06 | 3.8 |
| 4/2/2019 | 2400 | 0.04 | 1.7 | 0.44 | 1.34 | 0.08 | 67 |
| 7/16/2019 | 130 | 0.2 | 1.6 | 0.19 | 0.64 | 0.11 | 19.8 |

**Table A3.** IBWC North Floodway Water Quality.

| Date | Bacteria | Ammonia | TKN | TP | Nitrate + Nitrite | Chlorophyll-a |
|---|---|---|---|---|---|---|
| 11/3/2011 | 0 | 0.16 | 2.03 | 0.00 | 2.42 | 29.70 |
| 2/23/2012 | 0 | 0.09 | 0.95 | 0.21 | 5.28 | 35.00 |
| 5/3/2012 | 0 | 0.13 | 1.49 | 0.29 | 4.47 | 40.20 |
| 8/23/2012 | 0 | 0.12 | 1.04 | 0.23 | 2.26 | 55.70 |
| 11/19/2012 | 0 | 0.06 | 1.50 | 0.59 | 2.75 | 42.60 |
| 3/12/2013 | 110 | 0.16 | 1.08 | 0.00 | 2.68 | 40.50 |
| 8/21/2013 | 640 | 0.23 | 0.89 | 0.23 | 2.01 | 51.40 |
| 11/25/2013 | 7300 | 0.12 | 0.68 | 0.41 | 3.96 | 9.50 |
| 8/14/2014 | 0 | 0.06 | 1.70 | 0.00 | 2.03 | 82.30 |
| 11/24/2014 | 1100 | 0.11 | 1.36 | 0.34 | 3.82 | 44.40 |
| 2/25/2015 | 110 | 0.13 | 1.57 | 0.27 | 3.08 | 35.40 |
| 3/26/2015 | 0 | 0.25 | 1.66 | 0.35 | 6.71 | 26.00 |
| 8/26/2015 | 1400 | 0.12 | 1.84 | 0.32 | 3.10 | 60.20 |
| 8/27/2015 | 0 | 0.07 | 1.53 | 0.26 | 3.02 | 76.20 |
| 11/30/2015 | 610 | 0.19 | 3.19 | 0.25 | 4.98 | 23.40 |
| 5/4/2016 | 360 | 0.21 | 2.01 | 0.31 | 4.37 | 68.30 |
| 8/4/2016 | 0 | 0.00 | 0.00 | 0.27 | 2.08 | 20.10 |
| 11/2/2016 | 95 | 0.05 | 0.74 | 0.42 | 2.98 | 52.80 |
| 2/8/2017 | 0 | 0.08 | 1.72 | 0.39 | 4.29 | 11.00 |
| 5/3/2017 | 75 | 0.08 | 1.55 | 0.27 | 4.37 | 2.31 |
| 7/25/2017 | 120 | 0.05 | 0.00 | 0.25 | 1.07 | 19.60 |
| 11/29/2017 | 160 | 0.00 | 0.00 | 0.00 | 0.00 | 9.94 |
| 1/30/2018 | 20 | 0.16 | 0.00 | 0.29 | 3.80 | 6.91 |
| 4/18/2018 | 340 | 0.05 | 1.29 | 0.50 | 4.43 | 66.90 |
| 7/18/2018 | 96 | 0.05 | 2.30 | 0.39 | 2.36 | 78.10 |
| 10/16/2018 | 300 | 0.29 | 1.51 | 0.57 | 1.79 | 72.30 |
| 1/23/2019 | 200 | 0.10 | 1.03 | 0.35 | 4.67 | 28.60 |
| 4/16/2019 | 1600 | 0.05 | 1.03 | 0.24 | 2.65 | 36.30 |
| 11/7/2019 | 0 | 0.21 | 1.20 | 0.15 | 2.35 | 32.60 |

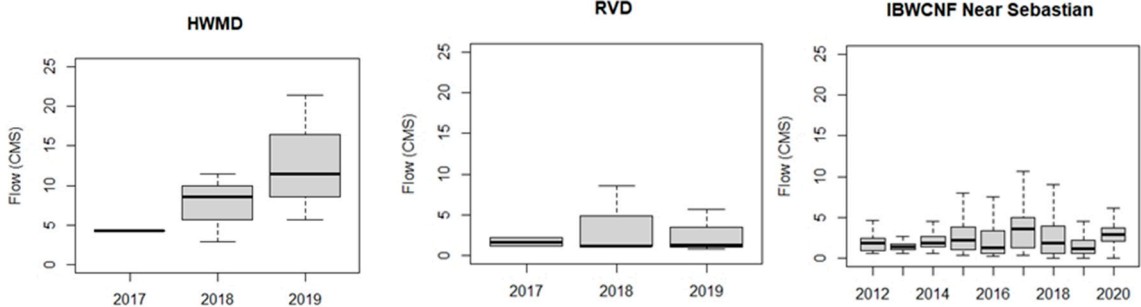

**Figure A1.** North and Central Watersheds Flow Data.

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
