# Peer review of "Development of a Cyberinfrastructure for Assessment of the Lower Rio Grande Valley North and Central Watersheds Characteristics"

_sustainability, doi:10.3390/su132011186_

Round 1
Reviewer 1 Report
This article analyzes the main sources of the impaired quality of the water of the Lower Laguna Madre watershed and its three mains subbasins. For that analysis, the authors perform the characterization of the basins by developing cyberinfrastructure and collecting a wide inventory of data to identify which one of the three subbasins has a major source of the water pollution into the Lower Laguna Madre basin.
General observations:
The results are not clearly presented. In many parts of this section is generated a discussion, which must be part of section 5. Namely, the authors mix the results with the discussion.
Lines 427, 552, 604, 646 shown the acronym TLPA, but these must be TLAP
In lines 38, 230, 234, 249 are shown bibliographies that are not listed in section references.
Material and Methods
Please provide an Annual Flow Histogram to check the flow variation throughout the year.
Results
Line 477. Please replace ”AS” by as
Line 498. Please replace “according to [39]” with “according to Olmstead [39]
Figure 5. Please, explain what is the meaning of the red line of the different graphs.
Discussion and Conclusion
Line 618: “Bacteria loadings were the only loading concentrations …” Please improve the redaction avoiding cacophonies.
Author Response
General observations:
- The results are not clearly presented. In many parts of this section is generated a discussion, which must be part of section 5. Namely, the authors mix the results with the discussion.
Response #1
The authors made a major change in the Results and Discussion sections, some of the sentences that were added initially to the Discussion were moved to the Results part. The Discussion part includes the outcomes and findings from the study without illustrations of parts related to the results.
Line numbers 399-417: “With the collaboration of REON, a cyberinfrastructure website, both data collection and the development of maps were accomplished. This platform provided an efficient watershed characterization by exposing significant guidelines from the EPA watershed characterization manual. This manual provides the basis to meet water quality and wa-tershed management goals. Physical and natural features, land use, waterbody condi-tions, pollutant sources, and waterbody monitoring information are the data needed to characterize a watershed [57]. The first step for the watershed characterization was to de-velop the watershed delineation for the three watersheds. The results were then uploaded to the REON website to show watershed boundaries. Additionally, NPS and PS pollution layers were included in each watershed to facilitate the characterization process based on EPA watershed characterization. The cyberinfrastructure gathers existing watershed boundaries, hydrology, land use, NPS pollution, PS pollution, water quality stations, and flow stations to support the overview of the watershed characteristics. Three maps were created: Watershed delineation results, NPS pollution, and PS pollution maps. The maps created facilitated the watershed characterization by integrating geospatial data for NPS and PS pollutants for each watershed individually. The development of maps portrayed in the cyberinfrastructure helped stakeholders collaborates in the characterization by providing inputs for each potential source that could contaminate the area. The web user interface at the regional level is available for every stakeholder no matter time or location.”
Line numbers 421-428: “The watershed delineation encompassed a comprehensive LIDAR elevation data reconditioning to well display the North and Central Watersheds’ characteristics. Elevation re-conditioning has revealed improved results in areas with very flat terrain. Previous studies had positive results with respect to their watershed delineation by performing this methodology [14]. Burning the waterways to the elevation data has enhanced the terrain to better support the current conditions of the elevation changes in the waterways. Generally, all the waterways within the area are man-made, which is challenging for the eleva-tion data to capture the waterways.”
Line numbers 460: This type of land use is within the downstream tributary areas of the watersheds.
Line numbers 503-507: “The identification of McAllen-Edinburg-Mission MSA in this watershed demonstrates the high presence of urban areas. The HWMD had 20.3% of urban areas and 8.8% from the three watersheds. In contrast, the IBWCNF presented a higher percentage of 24.3% in ur-ban areas, but it only had 5.8% with respect to the overall area of the North and Central Watersheds.”
Line numbers 511-514: “Major PS pollutants identified in this watershed were TLAP and MSW. The TLAP corre-sponds to the presence of high levels of nitrogen in the watershed and the MSW corre-sponds to the presence of high total phosphorus levels.”
Discussion and Conclusion section
Line numbers 661-668: The cyberinfrastructure and REON website contributed significantly to this study in portraying relevant characteristics of each of the North and Central Watersheds. The RE-ON website not only collects distinct information into one single source but also allows the stakeholders within each watershed to assess the watershed characteristics. Therefore, this platform is an innovative tool that supports effective watershed characterization. ArcGIS automated hydrology tools have shown to have satisfactory results in delineating watersheds. Overall, the study showed that the watershed delineation process used pro-vided acceptable results to characterize the North and Central Watersheds.
Line numbers 670-687: “Although the HWMD watershed was not the highest regarding the urban areas, it is considered higher in NPS pollution with respect to the entire area of the North and Central Watersheds. Urban areas have more impact on the HWMD in comparison to the other watersheds regarding the overall watershed areas. This finding suggests that urban areas in this watershed are linked to the presence of bacteria and chlorophyll-a. Based on the water quality data obtained, only chlorophyll-a levels were higher than the other water-shed levels. The high levels of chlorophyll-a relate to the HWMD watershed in extensive urban areas. Based on the total PS pollution found in the North and Central Watersheds, HWMD is the watershed to contribute a 3.66 ratio with respect to the watershed area. While this watershed has greater PS pollution than the other two watersheds, it is not par-ticularly the most affected watershed with respect to the drainage area. The NPS and PS results for HWMD were consistent with the elevated levels of the water quality data ana-lyzed from the SWQMIS database. Bacteria, total nitrogen, nitrate and nitrite, chloro-phyll-a, ammonia, total phosphorus, and organic nitrogen in HWMD had significant values in this watershed compared to the other watersheds. In addition, the high pollutant loadings in this watershed correspond to high flow values recorded. Therefore, more flow data is needed in the future to further support this characterization and make the proper connections between sources of pollution and pollutant loads.”
Line numbers 692-694: The RVD watershed pollutant loadings were generally low, but bacteria loadings were significant because of the high presence of NPS pollutants..
Lines 427, 552, 604, 646 shown the acronym TLPA, but these must be TLAP
Response #2
The authors modified the spelling for the “TLAP” based on the reviewer's comment.
Lines number 513: TLPA was changed to TLAP
Lines number 530: TLPA was changed to TLAP
Lines number 646: TLPA was changed to TLAP
Lines number 717: TLPA was changed to TLAP
Lines 38, 230, 234, 249 are shown bibliographies that are not listed in section references.
Response #3
The authors added the missing citations according to the Reviewers comment.
Line number 39: Texas Integrated Report is cited as citation [2]
Line number 266: Handbook for Developing Watershed Plans to Restore Our Waters is cited as citation [37]
Line number 269: 2016 National Land Cover Database is cited as citation [41]
Line number 325: the authors added the citation to the sentence is cited as citation [47]
Material and Methods: Please provide an Annual Flow Histogram to check the flow variation throughout the year.
Response #4
As suggested by the Reviewer, the authors added the figures below to show the variation of the flow data throughout the years for each waterway under supplementary materials in Appendix D.
Appendix D. North and Central Watersheds Flow Data
Line 477. Please replace ”AS” by as
Response #5
As suggested by the Reviewer, the authors revised the sentence as follows:
Line number 552: “In 2018, the TKN levels were the highest compared to the other years with more than 3.0 MGL as N.”
Line 498. Please replace “according to [39]” with “according to Olmstead [39]
Response #6
As suggested by the Reviewer, the authors revised the sentence as follows:
Line number 584: “The results suggest, according to Olmstead [47], that the watershed is affected by wildlife with small contributions of domestic animals and point sources.”
Figure 5. Please, explain what the meaning of the red line of the different graphs is.
Response #7
The authors agree with the Reviewer's comment. An explanation was added as follows:
Line number 546-548: “The water quality parameters samples for the north and central watersheds are shown in figure 6, where the red line represents the screening level according to TCEQ water quality standards.”
Line number 567: Redline represents the TCEQ screening level for each parameter
Line 618: “Bacteria loadings were the only loading concentrations …” Please improve the redaction avoiding cacophonies.
Response #7
The authors agree with the Reviewer’s comment and was modified as follows:
Line number 692-694: “The RVD watershed pollutant loadings were generally low, but bacteria loadings were significant because of the high presence of NPS pollutants..”

Reviewer 2 Report
Referee report
Manuscript: Sustainability-1333569
Title: Development of a Cyberinfrastructure for Assessment of the Lower Rio
Grande Valley North and Central Watersheds Characteristics
Authors: Linda Navarro, Ahmed Mahmoud*, Andrew Ernest, Abdoul Oubeidillah, Jessica Johnstone, Ivan Rene Santos Chavez, Christopher Fuller
The manuscript is interesting and fits well the scopes of Sustainability journal. It presents data concerning characteristics of the Rio Grande Valley Watersheds including river water degradation by anthropogenic pollution (non-point and point sources). Based on the national and state databases some maps of spatial distribution of the pollutants have been prepared. A few chemical and biological parameters have been presented. The discrimination between the sources from different origins has been carried out.
However, from the analytical point of view, the section 4.3 must be extended. Especially the description of methods (lines 274-276) must be given in some detail, or the procedures briefly summarized after quoting the pertinent literature concerning the applied methodologies. Moreover, interpretation of analytical results is in question. Concerning the chemical data elaboration, authors used only 8 samples for each watershed (please add a map with sampling sites). On the basis of such a small set of analytical data, it is not possible to reliably characterize the state of the environment in the area of 3,000 km2. I recommend reconsidering this interpretation.
Fig. 6 - are the concentration ranges selected automatically, randomly or do these ranges correspond to any classes of pollutants?
Please correct reference numbers: lines 53, 84, 87, 103, 128, 151, 153, 233, etc. - [3-5], [3,4]……
Author Response
The authors would like to thank the reviewer for his thoughtful comments and efforts towards improving the manuscript.
- However, from the analytical point of view, the section 3.3 must be extended. Especially the description of methods (lines 274-276) must be given in some detail, or the procedures briefly summarized after quoting the pertinent literature concerning the applied methodologies.
Response #1
As suggested by the Reviewer, the authors revised the sentences and updated the methods section as follows:
Under Methodology section:
Line numbers 233-238: “The watershed delineation process is fundamental for the overall characterization to define the watershed boundaries and subwatersheds within each watershed. Generally, the watershed slopes from west to east through the heart of the LRGV, with an average slope of fewer than 0.3 meters per kilometer [35]. Overall, its flat terrain varies from 0 m to 100 m. The resolution of the elevation raster-files was changed from 1 m to 60 m, which contributed to the reduction of file size and thus provided an efficient analysis.”
Line numbers 261-263: “The flow accumulation lines embody the actual waterways in mostly all the watersheds. The watershed boundaries correspond to the flowlines and follow an enhanced method-ology for the type of terrain in the region.”
Line numbers 277-290: “Cultivated crops and urban areas are two types of land cover that can be contributing to NPS pollution. Agricultural and stormwater runoff generated from cultivated crops and urban areas; respectively. Runoff carries various pollutants such as nutrients, sediments, heavy metals, and bacteria which has a negative impact on the receiving waterbodies [42]. In peri-urban areas, agricultural/rural NPS pollution and urban NPS pollution are two types of sources that have gained considerable concern because urban expansion and agriculture intensification may act as a source or sink for contaminants to move toward sur-face water bodies [43]. Agricultural and urban areas in a watershed have shown in previous studies to be the main contributors to NPS pollution. Another type of NPS pollutants source is the STLR. The main concern with this type of NPS pollutants is the exposure to several hazardous contaminants from the practice of livestock. The improper management of livestock wastes (manure) can cause surface and groundwater pollution [44]. Water pollution from animal production systems can be by direct discharge, runoff, and/or seepage of pollutants to surface or groundwater [45].”
Line numbers 291-313: “OSSFs are designed to treat domestic wastewater using a septic tank for screening and pretreatment and a drain field where pretreated septic effluent is distributed for soil infiltration and final treatment by naturally existing microorganisms [46]. Species with WMA were found close to the coast of each watershed. These NPS pollutants contribute to high bacteria loadings to waterbodies from wildlife in the region. Grazing animals and wildlife can also negatively affect the water quality of runoff and waterbodies with bacterial contamination [47]. In Texas, non-avian wildlife, such as deer or feral hogs, are commonly found to be significant contributors of bacteria to natural streams [44,47]. In addition, colonias are considered the most distressed areas in the United States. They are usually found along the U.S. - Mexico border, which often lacks necessities such as sewer systems, drinkable water, and overall sanitary housing. Many homes within colonias cannot meet county building codes because they lack indoor bathrooms and plumbing, a prereq-uisite for connection to local water lines and sewage systems [17]. Consequently, colonias can be a potential contributor of NPS pollutants since they lack adequate solid waste disposal and wastewater systems. TCEQ created a classification system to identify the colonias with adequate utilities and the ones that lack basic utilities. The red and yellow classification was the ones selected for colonias that potentially carry NPS pollution. Based on the priority classification by the Rural Community Assistance Partnership, OSSFs located in the colonias having a health hazard (red colonias) were assumed to have a greater failure rate of 70%. Conversely, a 30% failure rate (determined based on local expert knowledge) was assigned to areas having the lower priority ratings (yellow colonias) [47]. The term colonias refer to settlement or neighborhood that is unincorporated rural and peri-urban subdivisions along Texas' border with Mexico [48].”
Line numbers 334-354: “There is a substantial contribution of bacteria from wastewater outfalls, which potentially discharges to the waterways. Fecal contamination of water normally results from direct entry of wastewater from a municipal treatment plant into a water body [47]. There were two types of WWOs identified in these watersheds: domestic and industrial wastewater discharge. Domestic WWOs discharge less than 1 million gallons per day (MGD) while the ones with a discharge greater than 1 MGD may be either domestic sources or industrial wastewater treatment plant effluent. According to TCEQ, TLAP refers to the spreading of sewage from several applications, such as surface irrigation, evaporation, drain fields, or subsurface land application [50]. MSW facilities not only affect the surface water within the watershed but also groundwater. Closed landfills are commonly unlined and poorly capped, may be sources of a large number of organic compounds to surrounding groundwater and surface water [51]. Polluted stormwater runoff is commonly transported through MS4s, and then often discharged, untreated, into local water bodies [52]. MS4s are identified to discharge significant levels of contaminants to water-bodies in the United States and are now one of the major sources of water pollution in the nation [25]. Information about desalination plants were obtained from the Texas Water Development Board (TWDB) to support the PS pollution contribution to the watersheds. Disposing the concentrate from the desalination plant in the surface water is the most common method of concentrate disposal which is considered a point source [53]. These sources can be potential contributors to water quality impairments to the North and Central waterways.”
Line numbers 374-387: “Water quality data were obtained for the three watersheds from the Surface Water Quality Monitoring Information System (SWQMIS) database. The TCEQ maintains SWQMIS database to serve as a repository for surface water data throughout Texas. All the data available in the SWQMIS database has to be collected according to TCEQ surface water quality monitoring standards. Also, data must be verified and validated prior to its loading into SWQMIS. HWMD has a TCEQ monitoring station (ID 22003) located at FM 1420 1.65 KM south of the intersection with FM 490 east of Raymondville (Figure 3). Also, RVD has a TCEQ monitoring station (ID 22004) located at Willacy County Road 445 800 meters north of the intersection with FM 3142. Both HWMD and RVD monitoring stations have 8 water quality samples available on the SWQMIS database. Data from both sites were collected by Clean River Programs (CRP) from 2017 to 2019 [54]. For IBWCNF, one TCEQ monitoring station installed to collect water quality data since 2012. IBWCNF station ID is 20930 and located at US 77 2.5 KM south of the intersection of US 77 and FM 2629 in the city of Sebastian. There were 25 water quality samples for the IBWCNF water-shed available from SWQMIS from 2012 to 2019 [55,56].”
Moreover, interpretation of analytical results is in question. Concerning the chemical data elaboration, authors used only 8 samples for each watershed (please add a map with sampling sites).
Response #2
The authors used the data available collected by different agencies such as TCEQ and IBWC. The main issue that the three waterways was not fully characterized before this study and the efforts to collect samples had just started 2017 due to the apparent impact of the North and Central waterways on the Lower Laguna Madre which considered a unique estuary with social and economic benefits to the region. Especially with the rapid urbanization in the region that will have a greater effect on the water quality in the three watersheds in the future. Data availability was considered one of the limitations of the study. However, more data has to be collected in the future to validate the outcomes and findings for this study. But the results from this study should be the first step for the development of a watershed protection plan for each waterway which will serve as a strategy for effectively protecting and restoring aquatic ecosystem in the Laguna Madre. The authors added the following sentence:
Line number 723-726: “One of the limitations of this study was the acquisition of available data for such an ex-tensive study area of more than 3,000 km2. Additional flow data and water quality data could enhance the characterization as it was limited to only 8 samples for the HWMD and the RVD watersheds.”
On the same was A map was developed with the sample locations for the North and Central waterways as follows:
Line number 397:
Figure 3: Location for water quality and flow data monitoring stations.
On the basis of such a small set of analytical data, it is not possible to reliably characterize the state of the environment in the area of 3,000 km2. I recommend reconsidering this interpretation.
Response #3
The authors considered this interpretation as follows:
Line number 736-739: One of the limitations for this study was the acquisition of available data for such an extensive study area of more than 3,000 km2. Additional flow data could enhance the characterization as it was limited with only 8 samples for the HWMD and the RVD watersheds.
Fig. 7 - are the concentration ranges selected automatically, randomly or do these ranges correspond to any classes of pollutants?
Response #4
As suggested by the Reviewer, the authors clarified where the loadings were coming from.
Line number 635-637: “These loadings were generated automatically through ArcGIS properties to show the difference among pollutant loadings.”
Point #5
Please correct reference numbers: lines 53, 84, 87, 103, 128, 151, 153, 233, etc. - [3-5], [3,4]
Response# 5
The authors agree in modifying the reference numbers as mentioned by the Reviewer.
Line number 56: [3,4,5].
Line number 87: [4,13].
Line number 90: [4,14,15].
Line number 105: [1,20]
Line number 139: [27,28,29].
Line number 154: [27,28,29].
Line number 177: [32,33].
Line number 179: [1,33,34].
Line number 252: [36].
Line number 269: [38,39,40].
Line number 298: [44,47].
Line number 388: [55,56].
Line number 489: [51,52,53].

Reviewer 3 Report
This is a review for "Development of a Cyberinfrastructure for Assessment of the Lower Rio Grande Valley North and Central Watersheds Characteristics" by Navarro et al.
I appreciate the amount effort that went into the development of this work. It is well written and certainly the development of a large database describing water quality parameters of the LRGV would be immensely useful to stakeholders. Unfortunately, the work presents more like a technical report than a scientific study and I recommend it be rejected. In my opinion, there would have to be a substantial amount of work done on this manuscript to transform it into a scientific study.
I discuss only a few of the major issues below.
For example, in the Introduction there are few citations provided directly describing the LRVG region. Most citations are general in nature about watersheds and sources of pollution and not specific to this region. The motivation for this study is weak, stating only that the area is identified as an impaired water body (line 136).
There are no specific scientific goals stated. The paper is presented as designed to provide a comprehensive characteriation to analyze pollution sources (line 139) but no in-depth questions are really posed that the authors seek to answer and the motivation for this particular work is not well described. The authors state that "Quantifying this information will help identify" the primary source of water impairments but this language of "will help" is vague.
There is missing material from the methods section. For example, when describing the flow data, stations are identified but no further discussion is provided. Where are these stations? Are the stations representative of the watersheds?
In the results section (section 4.5 Flow data), the author's make unsubstantiated statments about the watersheds, for example "HWMD flow data reflects high flow values in 2019 with a mean value of 12 CMS. (lines 518-519). Why is this high? How do you know? What
is the variability of this data?
The discussion continues with "In 2018, the mean value was below 10 CMS. This level reflects high correlation with flooding patterns with respect to sudden storm events." (lines 519-520). Why is this true? No information about storm events has been provided.
This lack of detail impacts the author's discussion of Pollutant Loadings (section 4.6). The author's state "...loading calculations were obtained from quantifying flow and water quality data" (line 537). How exactly were the flow rates used? Since the flow rates are not presented with uncertainties, no determination can be made about the accuracy of the pollutant loadings.
When the loadings are discussed, vague quantifications are provided, for example "Bacteria unit area loading concentrations were determined to be high for the IWBCNF Watershed (lines 548-549). Why are these high? Relative to what standard?
The author's tend to use vague language or make speculations instead of being able to draw definative conclusions. For example:
(line 493) "The results suggest..."
(lines 508, 628) "This finding implies..."
(line 523) "...can potentially affect..."
and a longer example:
(lines 483-484) "The RVD Watershed had the higher levels of E. coli for the past 5 years compared to the other watersheds, which suggests that there are many septic tanks that can be leaking."
Are there other sources of E. coli that the authors can or have ruled out?
In summary, I appreciate the amount of work that went in to the creating of this work, but I do not believe it attains the standard of a scientific study.
Author Response
The authors would like to thank the reviewer for his thoughtful comments and efforts towards improving the manuscript.
- For example, in the Introduction there are few citations provided directly describing the LRGV region. Most citations are general in nature about watersheds and sources of pollution and not specific to this region. The motivation for this study is weak, stating only that the area is identified as an impaired water body (line 136).
Response #1
The motivation for this study was to characterize each North and Central Watershed that have not being characterized (EPA, 2017) and are considered to have insufficient data according to 2020 TCEQ Integrated Report (TCEQ, 2020). The reason that few citations provided directly describing the region, since no published data addressed the water quality issues in the North and Central Watersheds. Despite the fact this area is one of the fastest growing areas in the USA. So, this ongoing rapid urbanization within the region will increase the impervious surface. Therefore, more runoff will be generated and the Laguna Madre water quality will be greatly impacted. This study aims to quantify all the available geospatial data such as nonpoint and point sources through the cyberinfrastructure (Gutenson et al. 2020) and conduct an ample watershed delineation. Also, available water quality and flow data was incorporated to determine unit area pollutant loadings to have a better overview of which watershed will contribute the most to the Lower Laguna Madre impairments. Overall, study will support the development of a watershed protection plan for each North and Central watershed that will provide relevant information to contribute future studies.
Motivation:
- Cyberinfrastructure: Innovative method to obtain widespread of geospatial data for characterization incorporating utilization of information technology system in the watershed characterization process.
- First assessment for North and Central Watersheds which was not characterized that was not studied before, however, the water quality in the three waterways has an apparent impact on the Lower Laguna Madre.
- The Lower Rio Grande Valley is a unique floodplain study area which is distressed area that prone to get flooded:
- For example, Hurricane Hanna hit the LRGV in 2020 with 12-18 inches of rain in less than 48 hrs that caused severe damage for several areas across the region (Brownsville/RGV Forecast Office 2020)
- Heavy rain flooding event in 2018, most areas in the region received between 5 to 15 inches during 72 hrs period (Link)
- Determine the most affected watershed that contributes to the Lower Laguna Madre impairments. The Laguna Madre is located on the Texas coast, and it is the largest estuary and the southernmost. Additionally, it is one of only five hypersaline lagoons in the world. It is important for recreational and commercial activities, including marine transportation, oil and gas production, and business and residential development.
- The results from this study should be the first step for the development of a watershed protection plan for each waterway which will serve as a strategy for effectively protecting and restoring aquatic ecosystem in the Laguna Madre
There are no specific scientific goals stated. The paper is presented as designed to provide a comprehensive characterization to analyze pollution sources (line 139) but no in-depth questions are really posed that the authors seek to answer and the motivation for this particular work is not well described. The authors state that "Quantifying this information will help identify" the primary source of water impairments but this language of "will help" is vague.
Response #2
The main objective of this study is to characterize the North and Central watersheds that provide the Lower Laguna Madre with fresh water inflow. On the same time, the Laguna Madre is an impaired waterway and the sources of pollutants discharged to it are not fully studied and very limited information are available for each waterway. However, this study determined the main sources of pollution within each watershed through incorporating innovative technological systems such as cyberinfrastructure into the watershed assessment process. As suggested by the Reviewer, the authors clarified the statement as follows:
Line numbers 169-171: Quantifying this information will support the identification of which of the three watersheds contribute the most to water impairments in the Lower Laguna Madre by assessing each watershed independently.
There is missing material from the methods section. For example, when describing the flow data, stations are identified but no further discussion is provided. Where are these stations? Are the stations representative of the watersheds?
Response #3
The monitoring stations are managed by Texas Commission on Environmental Quality (TCEQ). Data collected from the monitoring stations are available on TCEQ Surface Water Quality Monitoring Information System (SWQMIS) database. The authors downloaded the data from the TCEQ database and compare it with the point and non-point source pollution identified on each watershed. Also, IBWC installed a flow monitoring station at the North Floodway, the authors used it also since it has a widespread flow observation at the location. As suggested by the Reviewer, the authors added more details on the monitoring station location used in the study and a map with the locations of the stations to support where the data is coming from. The flow data is reflected by the station of each waterway as explained in the following sentences:
Lines numbers 374-388: “Water quality data were obtained for the three watersheds from the Surface Water Quality Monitoring Information System (SWQMIS) database. The TCEQ maintains SWQMIS database to serve as a repository for surface water data throughout Texas. All the data available in the SWQMIS database has to be collected according to TCEQ surface water quality monitoring standards. Also, data must be verified and validated prior to its loading into SWQMIS. HWMD has a TCEQ monitoring station (ID 22003) located at FM 1420 1.65 KM south of the intersection with FM 490 east of Raymondville (Figure 3). Also, RVD has a TCEQ monitoring station (ID 22004) located at Willacy County Road 445 800 meters north of the intersection with FM 3142. Both HWMD and RVD monitoring stations have 8 water quality samples available on the SWQMIS database. Data from both sites were collected by Clean River Programs (CRP) from 2017 to 2019 [54]. For IBWCNF, one TCEQ monitoring station installed to collect water quality data since 2012. IBWCNF station ID is 20930 and located at US 77 2.5 KM south of the intersection of US 77 and FM 2629 in the city of Sebastian. There were 25 water quality samples for the IBWCNF water-shed available from SWQMIS from 2012 to 2019 [55,56].”
A map with the station’s location within the waterway of each watershed was added. The location of the stations is overlapping with each waterway representing the location for the data
Line numbers 396:
Figure 3: Location for water quality and flow data monitoring stations.
In the results section (section 4.5 Flow data), the author's make unsubstantiated statements about the watersheds, for example "HWMD flow data reflects high flow values in 2019 with a mean value of 12 CMS. (lines 518-519). Why is this high? How do you know? What is the variability of this data? The discussion continues with "In 2018, the mean value was below 10 CMS. This level reflects high correlation with flooding patterns with respect to sudden storm events." (lines 519-520). Why is this true? No information about storm events has been provided.
Response #4
The authors added the citations and more details on the flow data that support the high storm events as suggested by the reviewer, as follows:
Line numbers 600-611: “HWMD waterway flow data reflects high flow values in 2019 with a mean value of 12 CMS and in 2018 the mean value was below 10 CMS. These levels reflect a high correlation with flooding patterns with respect to sudden storm events from those years. Moreover, the RVD flow data showed high flow values in 2018 of almost 10 CMS (Appendix D). Both HWMD and RVD flow data correspond to past abnormal flooding events in the LRGV region. The region has experienced high storm events since 2018 with over 38.1 cm to 50.8 cm of rainfall causing severe flooding damage [69]. Such flooding's caused a halt to everyday functions for weeks and months because of minor to destructive varying degrees of flood damage in city roads, frontage roads, residences and businesses, and infrastructure in the LRGV region. Hidalgo, Cameron, and Willacy counties have received the Presidential Disaster Declaration in which have been determined to be the most impacted areas [70].”
In section 4.6, lacks of detail which impacts the author's discussion of Pollutant Loadings. The author's state "...loading calculations were obtained from quantifying flow and water quality data" (line 537). How exactly were the flow rates used? Since the flow rates are not presented with uncertainties, no determination can be made about the accuracy of the pollutant loadings.
Response #5
The authors calculated the pollutant loadings based on the guidelines of the United States Environmental Protection Agency (USEPA) Handbook for Developing Watershed Plans to Restore Our Waters. The following equation were utilized to calculate the loadings:
Pollutant Load Normalized by Area:
Total Pollutant Mass Normalized by Area:
where, where i = event 1,2,3...n
Ci = Concentration for event i = 1,2,3...n
A = Drainage Area
V = Event Volume
The flow rates from each waterway were used to calculate the unit area loading concentrations by multiplying the concentrations (water quality data) by the flow rate and divided by the watershed area. Then this unit loads were used to determine the pollutants load for each subwatershed from the North and Central Watersheds. The authors added a boxplot showing a summary of the data used in the study in Appendix D.
When the loadings are discussed, vague quantifications are provided, for example "Bacteria unit area loading concentrations were determined to be high for the IWBCNF Watershed (lines 548-549). Why are these high? Relative to what standard?
Response #6
The authors modified the sentence as suggested by the reviewer
Line numbers 637-646: “Bacteria loadings per unit area were determined to be slightly higher for the IWBCNF watershed than RVD. Despite IBWCNF has more potential NPS and PS sources for bacteria than RVD. The mean value for the bacteria loadings in IBWCNF and RVD was 12.4 (kg/km2/year) and 12.3 (kg/km2/year); respectively. This can be explained due to the main bacterial sources in both watersheds is coming from agricultural activities. The ratio in cultivated crops in IBWCNF was slightly higher than RVD. IBWCNF covered 59% of cultivated crops; while RVD covered 52%. Additionally, the flow volume in RVD was higher than IBWCNF. The average flow rate in RVD was 2.57 CMS; while in IBWCNF was 2.38 CMS. This could be the reason why the bacteria loadings in both watersheds have a minor difference”
The author's tend to use vague language or make speculations instead of being able to draw definitive conclusions. For example:
(line 493) "The results suggest..."
(lines 508, 628) "This finding implies..."
(line 523) "...can potentially affect..."
Response #7
The authors used selective speculative language only when necessary to make the argument more plausible. The word expressed in some sentences highlight certain findings to show the possibility of tested hypotheses which are beyond a strict understanding of the results. However, speculative language was based on evidence with an insightful interpretation that contribute to the advancement of knowledge in the North and Central watersheds characterization. The variety of phrases used in the study help the reader to distinguish between logical points and factual reporting of the outcomes that were extracted directly from the data analysis conducted in the study.
Lines 483-484. "The RVD Watershed had the higher levels of E. coli for the past 5 years compared to the other watersheds, which suggests that there are many septic tanks that can be leaking." Are there other sources of E. coli that the authors can or have ruled out?
Response #8
The authors provided more details to make the sentence clear.
Line numbers 569-572: “The RVD watershed had higher levels of E. coli compared to the other watersheds, which suggests that there could be several sources of NPS and PS such as septic tanks that can be leaking. Further, sewage may overflow from poorly structured sewage systems and create polluted stormwater runoff [65].”
The other sources are discussed under the Discussion and Conclusion.
Line numbers 690-693: “The results showed that the RVD Watershed has greater bacteria levels in comparison to the other watersheds, which suggests ranches and the activities within these areas are causing high levels of bacteria.”

Round 2
Reviewer 1 Report
The current structure of the article is much better.
Author Response
The authors would like to express their appreciation for the comments made by the reviewer which for sure improved the clarity and quality of the manuscript.
Reviewer 3 Report
I have read and reviewed the authors' responses to my initial review and am satisfied with their answers. I appreciate the significant effort made to improve the manuscript in response to comments from all the reviewers.
I believe the inclusion of the data and figures in the Supplementary material is necessary and very helpful.
Given their response to point 5, the manuscript would be improved if, at the very least, a citation was added after line 635 to their reference number 37 (USEPA Handbook) stating that the equations can be found in this exact reference, if the authors are not inclined to include them.
I don't agree with their response to point 7 where they argue to use speculative language when necessary to make the argument more plausible. Speculative language should be used to indicate to the reader where the scope of the definitive conclusions end.
Author Response
The authors would like to thank the reviewer for his thoughtful comments and efforts towards improving the manuscript.
Point #1
Given their response to point 5, the manuscript would be improved if, at the very least, a citation was added after line 635 to their reference number 37 (USEPA Handbook) stating that the equations can be found in this exact reference, if the authors are not inclined to include them.
Response #1
The authors added the following sentences as suggested by the reviewer:
Line 635-637:
Methods for calculating the loadings for each pollutant can be found in the USEPA Handbook for Developing Watershed Plans to Restore Our Waters [37].
Point #2
The author's tend to use vague language or make speculations instead of being able to draw definitive conclusions. For example:
(line 493) "The results suggest..."
(lines 508, 628) "This finding implies..."
(line 523) "...can potentially affect..."
I don't agree with their response to point 7 where they argue to use speculative language when necessary to make the argument more plausible. Speculative language should be used to indicate to the reader where the scope of the definitive conclusions ends.
Response #2
The authors appreciate your comments. The sentences were changed as follows:
Line numbers 584-585:
The results showed, according to Olmstead [47], that the watershed is affected by wildlife with small contributions of domestic animals and point sources.
Line numbers 587-588:
This finding indicates that the watershed is limited to carrying significant levels of ammonia from agricultural runoff.
Line numbers 595-596:
This finding indicates the presence of excess quantities of algae [68].
Line numbers 613-615:
Therefore, among the three watersheds, it has been determined that the HWMD waterway has the highest flow values that affect the loadings even if the water quality concentrations are low.
